# Structural basis for transcription antitermination at bacterial intrinsic terminator

Linlin You[1,2], Jing Shi[3], Liqiang Shen[1,2], Lingting Li[1,2], Chengli Fang[1,2], Chengzhi Yu[1,2], Wenbo Cheng[1,2], Yu Feng [3] & Yu Zhang [1]

Bacteriophages typically hijack the host bacterial transcriptional machinery to regulate their own gene expression and that of the host bacteria. The structural basis for bacteriophage protein-mediated transcription regulation—in particular transcription antitermination—is largely unknown. Here we report the 3.4 Å and 4.0 Å cryo-EM structures of two bacterial transcription elongation complexes (P7-NusA-TEC and P7-TEC) comprising the bacteriophage protein P7, a master host-transcription regulator encoded by bacteriophage Xp10 of the rice pathogen *Xanthomonas oryzae pv. Oryzae* (*Xoo*) and discuss the mechanisms by which P7 modulates the host bacterial RNAP. The structures together with biochemical evidence demonstrate that P7 prevents transcription termination by plugging up the RNAP RNA-exit channel and impeding RNA-hairpin formation at the intrinsic terminator. Moreover, P7 inhibits transcription initiation by restraining RNAP-clamp motions. Our study reveals the structural basis for transcription antitermination by phage proteins and provides insights into bacterial transcription regulation.

---

[1] Key Laboratory of Synthetic Biology, CAS Center for Excellence in Molecular Plant Sciences, Shanghai Institute of Plant Physiology and Ecology, Chinese Academy of Sciences, 200032 Shanghai, China. [2] University of Chinese Academy of Sciences, 100049 Beijing, China. [3] Department of Biophysics, and Department of Pathology of Sir Run Run Shaw Hospital, Zhejiang University School of Medicine, 310058 Hangzhou, China. Correspondence and requests for materials should be addressed to Y.F. (email: yufengjay@zju.edu.cn) or to Y.Z. (email: yzhang@sippe.ac.cn)

Bacteriophages are the most numerous and diverse organism life forms on earth[1]. They have emerged as attractive options for treatment of multi-drug resistant bacterial infection of human, for prevention and eradiation of microbial contaminant in food industry, and also for control of plant pathogenic bacteria[2–5]. Like other viruses, bacteriophages rely on host bacteria for reproduction and release of their progeny and therefore have evolved complex mechanisms for successful infection and propagation. One of the most frequently applied mechanisms is hijacking host bacterial transcription machinery to shift host resources towards their own need[6].

The bacterial transcription machinery is composed of a catalytic RNAP core enzyme with five subunits (2α, 1β, 1β′, and 1ω subunits), initiation σ factors responsible for precise initiation of RNA synthesis at promoter regions[7], various elongation factors responsible for efficient and coordinated extension of RNA molecules with high processivity and fidelity[8], and termination factors for stop of RNA synthesis at the end of operons[9]. During transcription initiation, σ factors engage with RNAP core enzyme to form a RNAP holoenzyme, which binds to and unwinds promoter DNA through a multiple-step process[7,10–13]. During transcription elongation, RNAP is under dynamic regulation by nucleic-acid signals and regulatory elongation factors, which in concert guide RNAP to proceed, pause, or backtrack[8]. Transcription is terminated mainly through two mechanisms in bacteria—the intrinsic termination (RNAP stops at a DNA sequence characterized by a G-C rich hairpin followed by a poly-T track) and the Rho-dependent termination (RNAP is dislodged by Rho factor)[9,14]. A recent report also highlighted an important role of NusA as a termination factor for preventing misregulation of global gene expression in *E. coli*[15]. The transcription initiation and termination are the most frequent targets of bacteriophage proteins during phage infection[6].

Xp10 is a virulent bacteriophage of *Xanthomonas oryzae pv. Oryzae (Xoo)*, a pathogenic bacterium of rice blight[16]. *Xoo* belongs to γ proteobacteria and thereby RNAP from this bacterium is similar to that of *E. coli*. The gene expression of phage Xp10 is under hierarchical temporal control by both host RNA polymerase and its own polymerase[16,17]. The transcription of early phage genes (encoding Xp10 RNA polymerase, transcription regulatory factor, and proteins involved for viral DNA replication, etc.) rely exclusively on host RNA polymerase at the early stage of infection. Subsequently, majority of host transcription is turned off ~10 min post Xp10 infection by inhibiting the host RNAP and the transcription of phage Xp10 late genes (encoding phage structural and host lysis proteins) is turned on[16–19].

P7 is the key phage protein to regulate gene expression programs of both host bacteria and phage Xp10. P7 binds to host RNA polymerase to inhibit transcription of host genes. Moreover, P7 also redirects the host RNAP to turn on transcription of phage late genes[20]. P7 inhibits host transcription initiation from σ[70]-driven promoters (both −35/−10 type and extended −10 type promoters) and σ[54]-driven promoters (−12/−24 type promoters)[16,20,21]. The molecular mechanism of transcription inhibition by P7 is unclear. Previous reports suggested that P7 might interfere with the engagement of σ factor with RNAP core enzyme based on the observation that preparation of *Xoo* RNAP from Xp10 phage-infected cells contains less σ factors compared with that from uninfected cells[19,22]. However, contradictory results showed that σ factors and P7 could bind to RNAP simultaneously in solution; instead the study showed that P7 probably inhibits RNAP-promoter open complex (RPo) formation through an unknown mechanism[23].

P7 turns on transcription of late genes of phage Xp10 by assisting host RNA polymerase in bypassing intrinsic terminators preceding the late genes[16,17,20]. The underlying mechanism of P7-mediated antitermination remains elusive. A recent report suggested that P7 might inhibit transcription pausing by promoting forward translocation, thus affecting the first step of intrinsic termination. However, the marginal effect of P7 on elemental transcription pause could not explain its strong inhibition on transcription termination[24]. It is intriguing that NusA, a transcription elongation factor that facilitates transcription termination, is reprogrammed into an anti-terminator by P7 and enhances P7-mediated transcription termination through an unknown mechanism[25].

It is very intriguing to understand how P7 is capable of inhibiting the two distinct stages of transcription—the initiation and termination. Structural and biochemical studies suggest that P7 interacts with the N-terminal short helix of RNAP-β′ subunit (β′ NTH)[23,26], the RNAP-β flap domain (βflap)[23], and RNAP-ω subunit[25]. Although the binary NMR structure of β′NTH-P7 provides details of interactions between P7 and β′NTH, it is difficult to estimate the binding site of P7 on RNAP, as the non-conserved β′NTH is disordered in all reported RNAP structures.

In this study, we determined a cryo-EM structure of *Xoo* transcription elongation complex with P7 (P7-TEC) at 3.95 Å and a cryo-EM structure of *Xoo* transcription elongation complex with P7 and NusA (P7-NusA-TEC) at 3.41 Å. The structures reveal that P7 plugs up the RNA exit channel of RNAP, where it is further stabilized by NusA. With such interaction, P7 restrains the RNAP motions and reshapes the RNA exit channel. The cryo-EM structures combined with biochemical data suggest mechanisms that P7 employs to modulate host RNAP—P7 inhibits transcription initiation by preventing opening of RNAP clamp for unwinding and loading promoter DNA into RNAP main cleft; and P7 impedes transcription termination by blocking RNA exit channel and preventing formation of the terminator hairpin.

## Results

**Structure determination of *Xoo* P7-TEC and P7-NusA-TEC.** To understand the structural basis for transcriptional regulation of host transcription machinery by the bacteriophage protein P7, we sought to obtain a structure of *Xoo* transcription elongation complex with P7 (P7-TEC) and a structure of *Xoo* P7-TEC with NusA (P7-NusA-TEC). P7 exhibits expected inhibitory activity on transcription initiation and transcription termination of reconstituted *Xoo* RNAP (Supplementary Fig. 1a–c). We prepared the complexes by direct reconstitution with *Xoo* RNAP core enzyme, P7, an elongation nucleic-acid scaffold, and NusA (Fig. 1a and Supplementary Fig. 2a–f). The purified complexes show equal molar ratio for RNAP core enzyme and the transcription factors (Supplementary Fig. 2a–f). The structures were determined to nominal resolutions of 3.95 Å for P7-TEC and 3.41 Å for P7-NusA-TEC by the single-particle cryo-EM method (Supplementary Table 1; Supplementary Figs. 2g–l and 3).

The overall structures of RNAP and the interactions between P7 and RNAP are essentially the same in the two structures (Supplementary Fig. 4a–e). Therefore, the 3.41 Å P7-NusA-TEC structure was used for discussion hereafter unless specified. The cryo-EM map of P7-NusA-TEC shows clear electron densities for 6-bp upstream double-stranded DNA (dsDNA), 9-bp DNA-RNA hybrid of the transcription bubble, 3-nt single-strand RNA (ssRNA) in the RNA exit channel, and 13-bp downstream dsDNA (Fig. 1b, d). More importantly, the cryo-EM map unambiguously shows the electron density for P7 at the RNA exit channel (Fig. 1b, e). The NMR structure of P7 (PDB: 2MC6) could be readily fit into the density with minor adjustment[23].

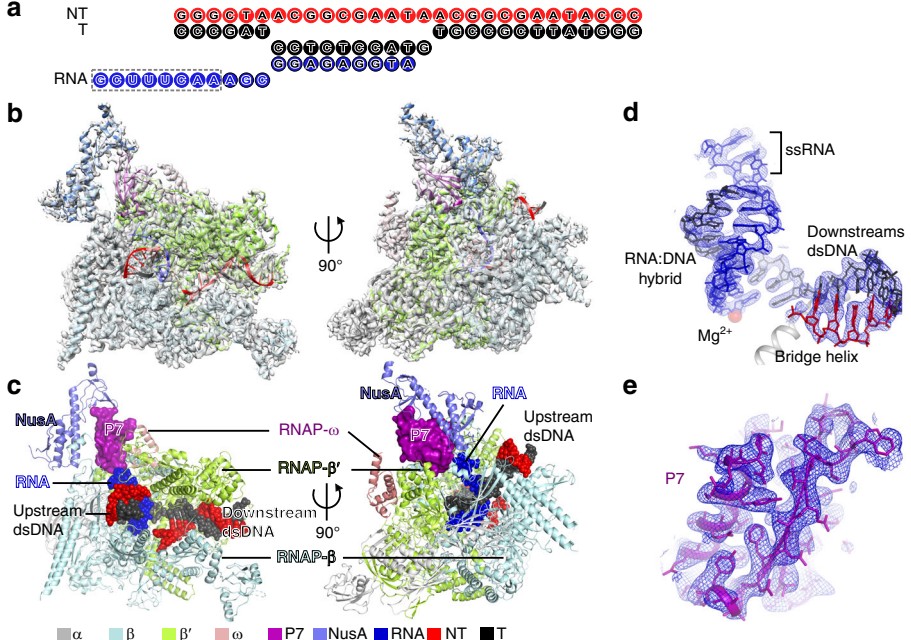

**Fig. 1** The overall structure of P7-NusA-TEC. **a** The scaffold used for obtaining the cryo-EM structures. Nucleotides in dashed box are disordered in the structure. **b** The map and model of P7-NusA-TEC in top and side views. The cryo-EM electron density map was shown as gray surface. **c** The overall structure of P7-NusA-TEC in top and side views. **d** The map and model show unambiguous density for 3-nt ssRNA in the RNA-exit channel and for RNA–DNA hybrid in a post-translocation state. $Mg^{2+}$ represents the catalytic magnesium ion at the active center. **e** The map and model for P7. The cryo-EM electron density map was shown as blue mesh

**P7 plugs up the RNA-exit channel of the host bacterial RNAP.** The RNAP clamp in the P7-NusA-TEC is closed and the overall conformation of *Xoo* RNAP is essentially superimposable to *E. coli* TEC (Supplementary Fig. 4f)[27]. The DNA–RNA hybrid in the P7-NusA-TEC adopts a canonical post-translocation conformation as in *E. coli* TEC (Fig. 1d and Supplementary Fig. 4e), suggesting that P7 does not affect the conformation of the DNA and RNA during processive transcription elongation. P7 inserts into the RNA exit channel and makes extensive interactions with RNAP (Figs. 1c and 2a) with calculated interface of 1182 Å[2]. P7 binds to RNAP core enzyme, RNAP holoenzyme, or TEC with *Kd* values of 6.1 nM, 11 nM, or 6.6 nM in a fluorescence polarization (FP) assay, respectively (Supplementary Fig. 1e–h), supporting the high affinity of P7 to RNAP and explaining that P7 was co-purified with RNAP in the Xp10 phage-infected cells[20].

The globular protein P7 comprises a β sheet at the N-terminus packed against the C-terminal α-helix (Figs. 1e and 2a). One side of the β sheet makes interactions with the N-terminal short helix of RNAP-β′ subunit (β′NTH; Fig. 2a). Such interactions have been mapped previously and the binary NMR structure of β′NTH-P7 is superimposable to the P7-NusA-TEC (Supplementary Fig. 5a)[23]. The interface is mainly composed of hydrophobic residues (I7, L22, F50, V51, and V54 of P7 and L4, L5, L7, and F8 of RNAP-β′ subunit; Fig. 2b and Supplementary Fig. 5b). Truncation of β′NTH (Δβ′NTH-RNAP) or alanine substitution of non-polar residues of P7 (F50A and V51A) causes substantial loss of antitermination effect of P7 (Fig. 2c), consistent with previous findings and suggesting a functional relevance of the interaction[23,26,28]. The other side of the β sheet approaches to but does not make interaction with the flap domain of RNAP-β subunit (βflap) in both P7-TEC and P7-NusA-TEC structures (Fig. 2a and Supplementary Fig. 4d).

The C-terminal helix of P7 (residues 48–62) forms a four-helix bundle with βCTR and β′dock (Fig. 2a, d). The interactions in the helix bundle are mainly mediated by non-polar residues including residues Y44, P48, V51, and A52 of P7, residues Y1347, V1351 of RNAP βCTR, and residue I394 of RNAP β′dock; three plausible H-bonds (β′E386 forms two H-bonds with S56 and R60 of P7, and β′K398 forms one H-bond with P7 residue E49) probably also contribute to the interactions (Fig. 2d, e, Supplementary Fig. 5c, d). Mutation of the interface residues greatly impaired the antitermination effect of P7, suggesting that such interface is also important for the engagement of P7 with RNAP (Fig. 2f). We did not observe interactions between the RNAP-ω subunit and P7 due to absence of density of the C-terminal half helix of RNAP-ω subunit in our structures (Supplementary Fig. 5e). Truncation of the C-terminal half-helix of RNAP-ω subunit or the C-terminal loop of P7 (the two regions might interact with each other in a structure model with extented RNAP-ω C-terminal helix; Supplementary Fig. 5e) has slightly alleviated the inhibitory activity of P7 on transcription termination (Supplementary Fig. 5f), but has no obvious effect on the action of P7 on transcription initiation, consistent with the previous study[25].

**P7 prevents RNA-hairpin formation in the RNA-exit channel.** Our cryo-EM structures show that P7 almost completely blocks the RNA-exit channel (Fig. 3a, b). The outer rim of the RNA-exit channel constitutes a narrow gate with a diameter of ~27 Å for exit of single-stranded RNA during processive transcription elongation (Fig. 3c)[29,30]; the gate can be further widened to a diameter of ~30 Å when invaded by an RNA hairpin (Fig. 3d)[31,32]. In the structures of P7-NusA-TEC and P7-TEC, such gate is largely blocked by P7 and leaves limited space for even a single-stranded RNA (Fig. 3b, e). Surface presentation of the structure suggests an alternative gate with a diameter of ~18 Å created by the βflap, β′ ZBD, and β′Zipper (Fig. 3f–g and Supplementary Fig. 5h). In the RNA exit channel, the residues of RNAP- β′subunit (M1 and K2) and P7 (R41 and D47) make direct contacts with RNA and guide the RNA towards the new gate (Supplementary Fig. 5i).

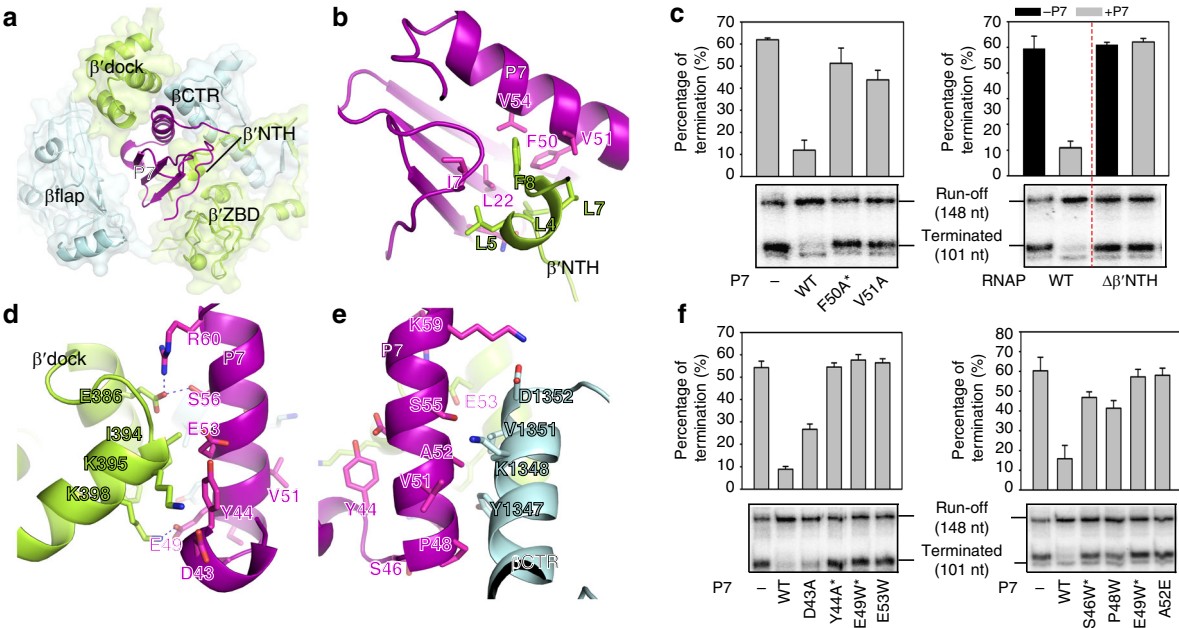

**Fig. 2** P7 plugs up the RNA-exit channel of RNAP. **a** The close-up view of the P7 binding pocket. **b** The detailed interactions between P7 and RNAP β′NTH. **c** The in vitro transcription results showing that disrupting the P7/β′NTH interface affects antitermination effect of P7. The area of terminated transcript contains two RNA bands, of which only the upper one is affected by P7 and thereby quantitated. **d** The detailed interactions of P7 and RNAP β′dock. **e** The detailed interactions of C-terminal helix of P7 and RNAP βCTR. **f** The in vitro transcription results showing that disrupting the P7/β′dock or P7/βCTR interfaces affects antitermination effect of P7. The in vitro transcription experiments were repeated in triplicate, and the data were presented as mean ± S.E. M. The source data of Fig. 2c, f are provided in the Source Data file. The asterisks in **c** and **f** indicate substitutions that might affect the overall structure of the P7 mutants as visible in the CD spectra shown in Supplementary Fig. 5k

The structure of P7-TEC graphically signifies that P7 inhibits intrinsic termination by preventing the second step—formation of the termination RNA hairpin. The intrinsic terminator is characterized by a G-C rich hairpin followed by a U-track. Transcription termination at intrinsic terminators contains three successive steps—pause at the U-track, hairpin nucleation, and hairpin completion[9]. It is suggested that the terminator hairpin invades into the RNA exit channel during transcription termination by using a similar mechanism as in hairpin-dependent transcription pausing. Although no structural information is available for transcription termination[9], superimposition of P7-NusA-TEC onto the structure of an *E. coli* hairpin-dependent paused elongation complex (PDB: 6ASX) clearly shows a severe steric clash between P7 and the RNA hairpin (Fig. 3e), suggesting that nucleation of hairpin is not able to occur at the original RNA exit gateway[31]; moreover, the proposed alternative RNA exit gate is too narrow to allow nucleation of RNA hairpin (Fig. 3f–g). The much narrower alternative RNA gate in the P7-NusA-TEC should interact with the ssRNA more tightly than the regular one and thus might account for the effect of promoting forward translocation by P7[24].

To further validate our hypothesis, we performed in vitro transcription assay using promoter DNA with sequences of elemental pause (without pause hairpin) and hairpin-less tR2 terminator (Supplementary Fig. 1d). Transcription pauses at elemental-pause and poly-U sites, but P7 shows marginal effect on both pause efficiency and kinetics of pause escape (Fig. 3h). However, P7 substantially inhibits the pause efficiency at the hairpin-dependent pause site and almost completely inhibits transcription termination at the tR2 terminator (Fig. 3i and Supplementary Fig. 1d), in agreement with the hypothesis that P7 prevents hairpin formation either in hairpin-dependent pause or intrinsic transcription termination. Moreover, P7 substantially inhibits the binding of a *his*-pause nucleic-acid scaffold comprising an RNA hairpin to RNAP core enzyme but has no effect on the binding of a nucleic-acid scaffold without RNA hairpin, consistent with the previous finding that P7 itself has little effect on poly-U pause and further supporting the hypothesis (Fig. 3j, k and Supplementary Fig. 5j)[25].

**NusA interacts with P7 to enhance its antitermination activity.** NusA functions as an elongation factor, which stabilizes hairpin-dependent pause and promotes intrinsic transcription termination[15,33,34]. However, previous work reported that NusA functions as an antitermination factor in the presence of P7[25]. Our structure of P7-NusA-TEC provides structural explanation how P7 reprograms NusA to enhance its own antitermination activity. In the structure, NusA locates at outside of RNA exit channel (Fig. 1c), where it interacts with the tip helix of βflap domain (βFTH) with a hydrophobic groove of its NTD domain (Fig. 4a, b and Supplementary Fig. 6c). The conformation and binding location on RNAP of NusA in our structure of P7-NusA-TEC is similar to that of the NusA in the structure of *E. coli* NusA-PEC[32], but distinct from that of the NusA in the structure of λN-mediated transcription antitermination complex[35,36] (Supplementary Fig. 6b).

The structure explains how P7 reprograms NusA into an antitermination factor. P7 resides between NusA and the RNA exit channel of RNAP (Fig. 1c); the β2/β3 loop of P7 inserts into a cavity of NusA S1 domain and creates a moderate interface of 221 Å² (Fig. 4c and Supplementary Fig. 6d). Specifically, the W32 of p7 inserts into a phenylalanine cage of F155, F165, and F200 on NusA (Fig. 4c and Supplementary Fig. 7d). The S1 domain of NusA has been suggested to interact with RNA hairpin in the RNA exit channel thus stabilizing the hairpin-dependent paused elongation complex (PEC)[32]. Structure superimposition between the *Xoo* P7-NusA-TEC and *E. coli* NusA-PEC reveals that the

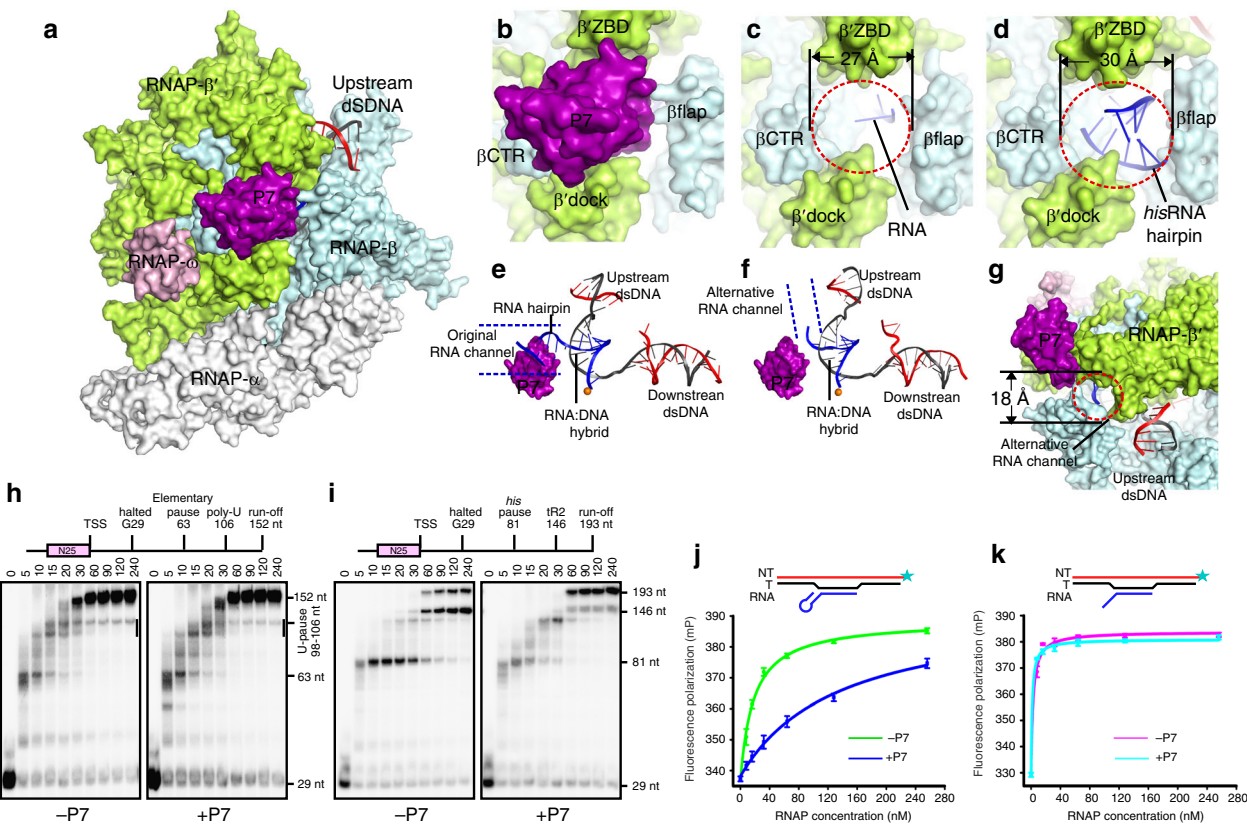

**Fig. 3** P7 inhibits pausing and intrinsic termination by preventing formation of RNA hairpin in the RNA-exit channel. **a** The surface presentation of P7-NusA-TEC shows the binding site of P7 on RNA polymerase. **b** The close-up view of P7 and its binding site. **c** The RNA-exit channel and RNA path in *E. coli* transcription elongation structure (PDB: 6ALF). The RNA-exit channel is in surface presentation and the RNA (blue) is shown as cartoon. **d** The RNA-exit channel and RNA hairpin (blue) in the *E. coli* paused elongation complex (*Ec* PEC; PDB: 6ASX). The RNA-exit channel is in surface presentation and the RNA hairpin (blue) is shown as cartoon. **e** Structure superimposition of *Xoo* P7-NusA-TEC and *Ec* PEC shows a steric clash between P7 and RNA hairpin in the RNA exit channel. **f** P7 guides the RNA to exit through an alternative gate. **g** The surface presentation showing the alternative RNA-exit gate in the P7-NusA-TEC structure. **h** P7 shows marginal effect on transcription pauses at elemental and poly-U pause sequences. **i** P7 substantially inhibits hairpin-dependent pause and intrinsic termination. **j** P7 significantly inhibits RNAP from binding a nucleic-acid scaffold with an RNA hairpin. **k** P7 has no effect on binding affinity of RNAP to an elongation nucleic-acid scaffold without RNA hairpin. The experiments were repeated in triplicate, and the data were presented as mean ± S.E.M. The source data of Fig. 3h–k are provided in the Source Data file

hairpin-interacting regions of NusA has been occupied by P7 (Fig. 4d), explaining that NusA losses its activity of enhancing pause or intrinsic termination in the presence of P7[25]. Instead, the interactions of NusA and P7 would stabilize P7 in the transcription elongation complex, providing the structural explanation for the increased affinity of P7 to RNAP and enhanced P7 efficiency of antitermination in the presence of NusA[25].

The interactions of NusA and P7 would further stabilize the closed conformation of RNAP induced by P7 (discussed below) and consequently prevent backtracking, probably accounting for reduced pausing at poly-U site in the presence of both NusA and P7[25].

**P7 inhibits transcription initiation by jamming the RNAP clamp.** Our structures also reveal the structural basis for inhibition of transcription initiation by P7 (Supplementary Fig. 1a). Disrupting the interactions of P7-β′NTH, P7-βCTR, or β′ dock, which are observed in the structure of P7-NusA-TEC, greatly alleviated inhibition of transcription initiation by P7 (Supplementary Fig. 7a), consistent with previous observations and indicating that P7 binds to the same site as in TEC to inhibit transcription initiation[23].

In the structure of P7-NusA-TEC, the RNAP clamp (a rigid domain spanning ~70 Å in distance from RNA-exit channel to

the dsDNA channel of RNAP and including β′ZBD, β′zipper, β′ lid, β′CC, β′rudder, and a helix-bundle; Supplementary Fig. 7b) adopts the closed conformation (Fig. 5a and Supplementary Fig. 4f), which has been observed in *E. coli* TEC structure containing similar nucleic-acid scaffold[27]. Superimposition of the P7-NusA-TEC to bacterial RNAP structures with open clamps (a *M. tuberculosis* RNAP complexed with lipiarmycin and a *T. aquaticus* RPi) suggests that opening the RNAP clamp involves a ~20° swinging of the clamp domain and a large concomitant movement of its RNAP-β′ZBD domain (Fig. 5a and Supplementary Fig. 7c)[37,38]. Such movement of RNAP-β′ZBD domain would introduce a steric clash between RNAP-β′ZBD/β′NTH and P7 (Fig. 5b–e and Supplementary Fig. 7d). Conversely, P7 binding would lock the RNAP clamp in the closed conformation and prevents it from opening (Fig. 5b, c). Therefore, we propose that P7 probably inhibits transcription initiation by jamming the RNAP clamp.

In agreement with the structure prediction, P7 barely binds to *Xoo* RNAP holoenzyme in the presence of lipiarmycin, an RNAP inhibitor which has been demonstrated to lock the RNAP in an open-clamp conformation (Fig. 5f)[37,38]; while Rifampicin, another RNAP inhibitor tolerant both clamp conformations, shows no effect on P7 binding (Fig. 5f)[39]. Because promoter melting during RPo formation in the stage of transcription

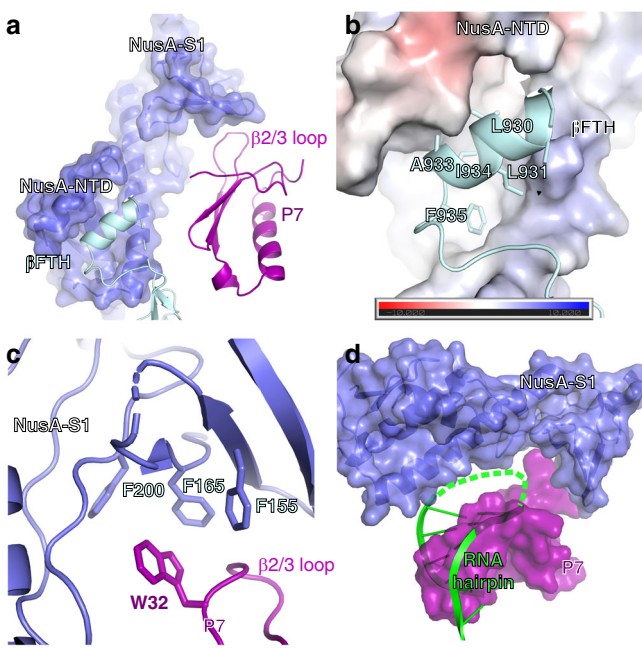

**Fig. 4** NusA interacts with P7 to enhance its antitermination effect. **a** NusA interacts with both RNAP βFTH and P7. **b** NusA interacts with the RNAP βFTH through a hydrophobic groove on its NTD domain. The electrostatic potential surface of NusA was generated using APBS tools in Pymol. **c** The detailed interactions between NusA and P7. **d** Structural superimposition between *E. coli* NusA-PEC (PDB: 6FLQ) and *Xoo* P7-NusA-TEC suggests that P7 blocks the RNA-hairpin binding site of NusA

initiation necessitates a flexible RNAP clamp[13,39–41], we thereby predict that P7 should inhibit melting and loading of promoter DNA into RNAP main channel. Supporting the hypothesis, P7 almost completely inhibits binding of promoter dsDNA to RNAP, but only slightly affects binding of pre-melted DNA scaffolds (Fig. 5g, h). To directly demonstrate that P7 inhibits promoter melting during RPo formation, we adapted a stopped-flow fluorescence assay, in which the promoter melting process could be monitored in real-time by the increase of fluorescence of a Cy3 probe attached to the nontemplate+2 nucleotide[40]. The results showed that both P7 (locking the clamp in the closed conformation) and lipiarmycin (locking the clamp in the open conformation) substantially inhibits the melting of promoter DNA, highlighting the importance of clamp mobility during RPo formation (Fig. 5i). Such clamp-jamming mechanism of P7 also explains its broad inhibition of transcription from σ70 and σ54-driven promoters[21]. The P7-induced closed conformation of the RNAP clamp also explains the effect of promoting forward translocation by P7 during transcription elongation[24].

The structure of P7-TEC also reveals an incompatible binding of P7 and domain 4 of σ70 (σ70₄) to RNAP (Fig. 6a). P7 makes interactions with the β′dock, β′ZBD, as well as βCTR, all of which are structural elements required for σ70₄ to anchor RNAP core enzyme (Fig. 6b, c)[42–46]. Furthermore, the flap-tip helix βFTH—the most important anchor point on RNAP for σ70₄—is displaced by P7 and becomes disordered in the structure of P7-TEC (Fig. 6e and Supplementary Fig. 4d). Superimposition of *Xoo* P7-TEC and *E. coli* RPo demonstrates that P7 and σ70₄ bind to the same site on RNAP core enzyme in a mutually exclusive manner (Fig. 6d), explaining that P7 slightly destabilizes RNAP holoenzyme[22,23].

## Discussion

In this work, we have determined a cryo-EM structure of *Xoo* P7-TEC at 3.95 Å and a cryo-EM structure of *Xoo* P7-NusA-TEC at

3.41 Å. The structures reveal a unique binding site (the RNA-exit channel) of the phage protein P7 on bacterial host RNAP, where P7 exerts dual regulatory activities on host RNAP—inhibiting transcription initiation and preventing transcription termination. The structures also unveil mechanisms of regulation on bacterial host RNA polymerase by phage proteins. P7 substantially narrows the RNA exit channel and consequently prevents formation and invading into the original RNA exit channel of RNA hairpin, providing structural basis for its antitermination activity (Fig. 7a). Moreover, P7 restrains RNAP motions and prevents opening of the RNAP clamp, consequently inhibiting RNAP from loading and melting promoter DNA (Fig. 7b).

Intrinsic terminator encodes a GC-rich terminator hairpin immediately followed by an T-rich tract[47–49], both of which are necessary for efficient transcription termination[9,14]. RNAP first pauses at the end of U-rich tract, providing a time window for the nucleation of the terminator hairpin[50,51]. The subsequent completion of termination hairpin unwinds the first 2–3 bp of DNA-RNA hybrid causing rearrangement of nucleic-acid scaffold and RNAP, which commits the elongation complex to termination pathway[52–55].

Our structures suggest that P7 inhibits the intrinsic termination by preventing the step of RNA-hairpin nucleation (Fig. 7a). Although no structural information is available for any intermediate states of transcription termination, it is proposed that a large RNAP conformational change occurs while formation of the partial terminator hairpin in the RNA exit channel[27,29]. The available cryo-EM structures of *E. coli* paused TEC and *E. coli* NusA-stabilized paused TEC with a RNA hairpin in the RNA exit channel revealed that the invading hairpin widens the RNA exit channel and induces a 'swivel' conformation of RNAP[31]. Our structures show that P7 almost completely blocks the RNA exit channel (Fig. 3a); such interactions leave no room for nucleation and invasion of RNA hairpin into the channel and accordingly should inhibit any events required for hairpin-formation in the RNA exit channel. In agreement with the prediction, P7 substantially inhibits the hairpin-dependent *his* pause while has little effect on the elemental or poly-U pause (Fig. 3f)[25]. Our structures support the proposal that the hairpin invasion is the obligatory step for intrinsic termination[9].

The bacteriophage protein N (phage λ), Q (phage λ, phages φ21, and φ84), and gp39 are other reported examples of antitermination factors[6,56]. A recent report of cryo-EM structure of λN antitermination complex suggests that λN forms a large complex with *nut* RNA and host elongation factors (i.e. NusA, NusB, NusE, and NusG), which shields the RNA-exit channel[36]. λN induces a large conformational change of NusA and creates a continuous surface; such interface potentially could trap the emerging RNA and thus prevent hairpin formation[36]. The recent high-resolution cryo-EM structure of λN-TAC reveals that the C-terminal loop of λN penetrates the RNA exit channel and probably compete with RNA-hairpin formation in the channel[35]. Extensive biochemical evidence suggests that Q also locates near the RNA exit channel to prevent transcription termination, but the structural basis is unknown[57–59]. The gp39 also resides at outer rim of the RNA exit channel by interacting with both σA₄ and RNAP-β flap domain. Although the structure of gp39-engaged *T. thermophilus* RNAP holoenzyme could not explain how gp39 inhibits transcription termination, subtle movement of gp39 induced by dissociation of σA would possibly bring gp39 closer to the RNA exit channel and consequently interfere with RNA-hairpin formation in the channel[60–63]. Our structures provide structural explanation how inhibition of hairpin formation is achieved by the phage regulatory factor P7 (Fig. 7a). As all of the reported transcription antitermination factors (λN, Q, and gp39) locate at

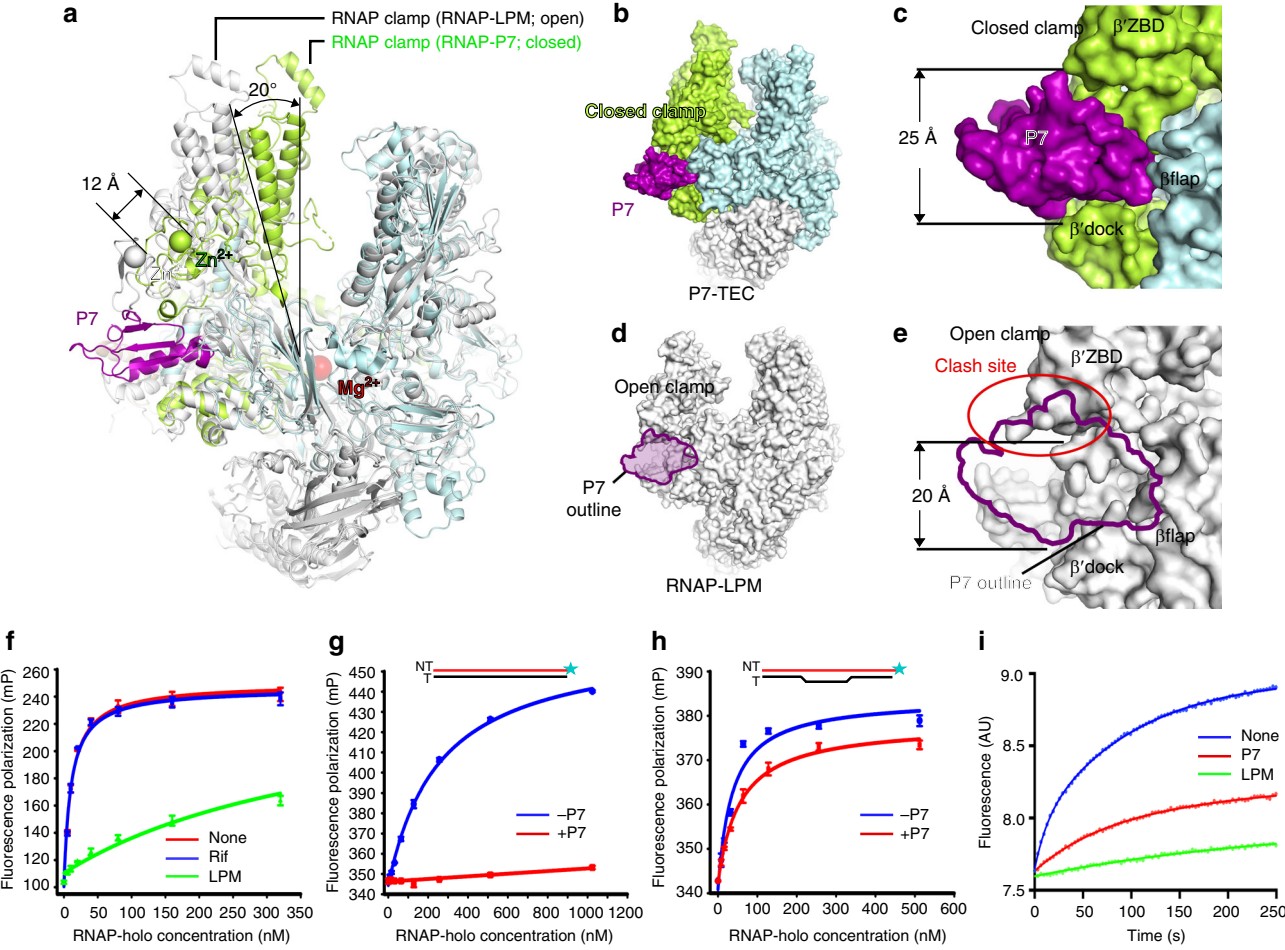

**Fig. 5** P7 locks RNAP clamp in the closed conformation. **a** Structural superimposition between *Mtb* RNAP-Lipiarmycin (PDB: 6FBV) with P7-NusA-TEC shows a 20° swinging of the clamp domain and a concomitant 12-Å movement of β′ZBD from the closed (P7-NusA-TEC) to open (RNAP-LPM) conformation of the clamp domain. **b** P7 inserts into the RNA exit channel and locks the clamp in the closed-conformation in the structure of P7-NusA-TEC. **c** The close-up presentation of the P7 binding site with a diameter of 25 Å. **d** P7 is not compatible with the open-clamp conformation. The P7 outline superposed onto the structure of *Mtb* RNAP-Lipiarmycin highlights the crushed P7 binding site in the open-clamp conformation. **e** The 12-Å shift of β′ZBD crushes into the P7 binding site and reduces the P7 binding site to a diameter of 20 Å. **f** Lipiarmycin (LPM) significantly inhibits the binding of P7 to RNAP, while rifampicin shows no effect in a FP assay. **g** P7 almost completely inhibits promoter dsDNA binding and isomerization by RNAP in a FP experiment. **h** P7 has little effect on the binding of pre-melted promoter of RNAP. **i** P7 inhibits promoter melting in a real-time stopped-flow assay. The experiments were repeated in triplicate, and the data were presented as mean±S.E.M. The source data of Fig. 5f–i are provided in the Source Data file

sites near the RNA exit channel, they probably prevent transcription termination through a unified mechanism.

During transcription initiation, RNAP-σ holoenzyme first loads onto promoter DNA and subsequently unwinds the promoter DNA starting from the −10 element. A large collection of single-molecular FRET, biochemical, and structural evidence demonstrated that successful promoter loading and unwinding require swinging motions of the RNAP clamp[13,39–41]. It is proposed that the RNAP clamp remains predominantly open for scanning in the genome, closes transiently to nucleate the promoter melting at the −10 element[40], reopens twice to sequentially clear two physical barriers (the β-lobe-gate barrier and FL2-Sw2 barrier) for loading promoter DNA into the RNAP main cleft[13], and finally form a catalytic-competent RPo[39,40,43].

RNAP clamp motions have been targeted by a few small-molecular-weight inhibitors of bacterial RNAP. Myxopyronin (Myx), corallopyronin (Cor), and ripostatin (Rip) interact with the RNAP "switch region"—the hinge of RNAP clamp motions—to prevent clamp opening and inhibit the isomerization of RPc to RPo (Supplementary Fig. 7f)[39,64,65]. Lipiarmycin binding at the base of RNAP clamp makes interactions with the other side of the

hinge to lock the clamp in an open conformation and inhibits its closure (Supplementary Fig. 7g)[37,38]. RNAP clamp motions have also been targeted by bacterial elongation factors. NusG interacts with both RNAP-β′ clamp and RNAP-β pincer to bridge the main cleft of RNAP, maintains a closed clamp conformation, and thus stimulates transcription elongation (Supplementary Fig. 7f)[30]. The phage protein Gp2, which inhibits host transcription, is also able to shift the equilibrium of clamp conformation towards a closed state (Supplementary Fig. 7f), but its main mechanism is to prevent the normal egress of σ70$_{1.1}$ from RNAP main cleft[66].

Our study provides an example of phage regulatory protein, which jams RNAP to inhibit bacterial transcription initiation through binding at a location distinct from those of any reported RNAP inhibitors and regulatory proteins (Fig. 7b and Supplementary Fig. 7f, g). P7 plugs up the RNA exit channel and makes extensive interactions with multiple structural motifs including essential interactions with the β′NTH of the closed RNAP clamp. The presence of P7 poses a steric hindrance for the movement of the β′NTH as well as β′ZBD of RNAP clamp and consequently locks the RNAP clamp in the closed conformation. The novel mechanism explains its broad inhibition on transcription

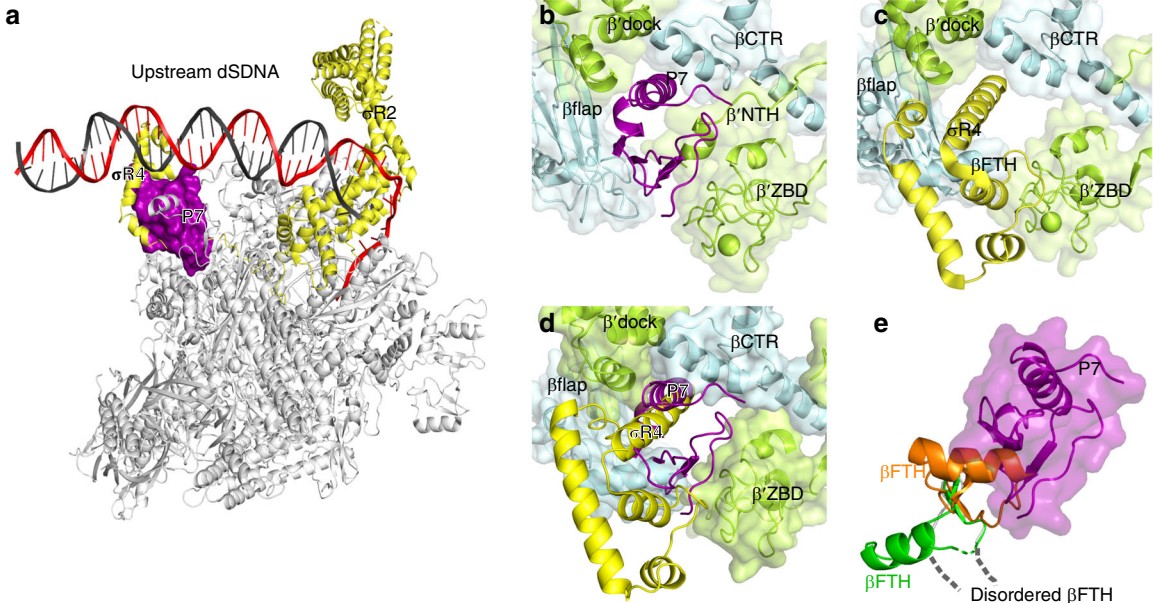

**Fig. 6** P7 displaces σR4. **a** The structural superimposition between *E. coli* RPo (PDB: 6CA0) and P7-TEC highlights the steric clash between P7 and σR4. **b** P7 binds to the outer rim of and inserts into the RNA-exit channel in the P7-TEC structure. **c** σR4 also binds to the outer rim of and inserts into the RNA-exit channel in the structure of *E. coli* RPo (PDB: 6CA0). **d** Both P7 and σR4 interact with the same structural modules in RNAP and are incompatible with each other. **e** Structural superimposition suggests steric clash between the RNAP βFTH and P7. The βFTH adopts different conformations in structures of P7-TEC (gray; disordered), P7-NusA-TEC (green), and RPo (orange)

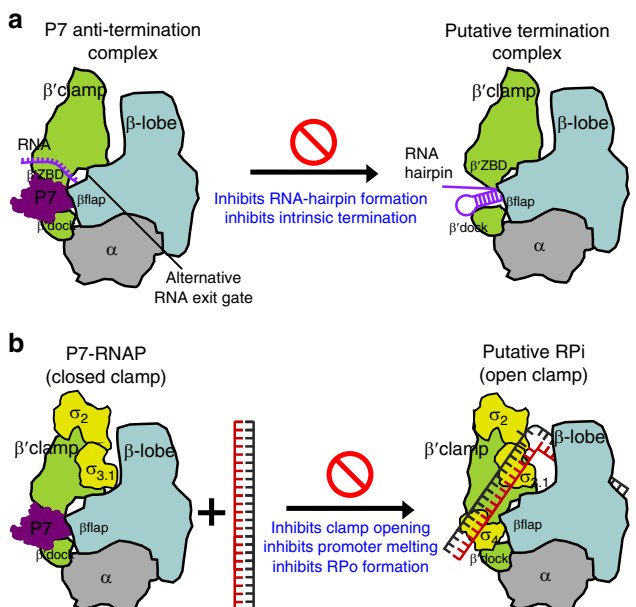

**Fig. 7** Proposed mechanisms for transcription inhibition and antitermination by P7. **a** P7 inserts into the RNA exit channel, re-directs RNA to exit in an alternative gate, and subsequently inhibits hairpin formation during hairpin-dependent pause and intrinsic termination. **b** The isomerization of dsDNA requires opening RNAP clamp during RPo formation. P7 inhibits transcription initiation by locking RNAP clamp in the closed conformation and consequently inhibiting unwinding and loading promoter DNA into the RNAP main cleft

initiation from all type of bacterial promoters (Supplementary Fig. 1a)[21]. Our study demonstrates that opening of RNAP clamp is an obligatory step in transcription initiation, which enables loading of promoter DNA into the RNAP active-center cleft during the isomerization of RPc to RPo[13,37,40,41,43,64].

Our cryo-EM structures and biochemical results provide the structural basis and molecular mechanisms for transcription inhibition and antitermination by the bacteriophage protein, P7. The binding site of P7 might serve as a new pocket for development of novel bacterial RNA polymerase inhibitors.

## Methods

**Plasmid construction**. Please see Supplementary Table 2 for the list of plasmids and Supplementary Data 1 for the list of primer information used in this study. The pET-28a-TEV-*Xoo* σ70 was constructed by inserting the *Xoo* σ70 gene amplified from *Xoo* genomic DNA into the pET-28a-TEV plasmid using NcoI and PstI restriction sites. The pTolo-EX5-*NusA* was constructed by inserting the *Xoo* NusA gene amplified from *Xoo* genomic DNA into pTolo-EX5 plasmid (Tolo Biotech.) using NcoI and PstI restriction sites. The pCOLA-*Xoo rpoB-rpoC* was constructed by inserting the *Xoo* rpoB, rpoC genes amplified from *Xoo* genomic DNA into the pCOLA-Duet (Merk Millipore) using NcoI/HindIII, NdeI restriction sites, respectively. The pACYC-*Xoo rpoA-rpoZ* was constructed by inserting the *Xoo* rpoA, rpoZ genes amplified from *Xoo* genomic DNA into pACYC-Duet plasmid (Merk Millipore) using BamHI/HindIII, and NdeI/KpnI restriction sites, respectively. The pET28a-TEV-P7 was constructed by inserting the synthesized DNA fragment (Genscript, China) into pET-28a-TEV using NcoI and PstI restriction sites. The pCOLA-*Xoo rpoB-rpoC* and pACYC-*Xoo -rpoA-rpoZ* derivatives were prepared by the same procedure. The pET28a-P7 derivatives were generated through site-directed mutagenesis (Transgen biotech, Inc.). The derivatives of pARTaq-N25-100-TR2 for in vitro transcription assays were obtained by site-directed mutagenesis.

**Protein preparation**. The *Xoo* RNAP core enzyme was prepared from *E. coli* strain BL21(DE3) (Novo protein, Inc.) transformed with plasmids pCOLA-*Xoo rpoB-rpoC* and pACYC-*Xoo rpoA-rpoZ*. Protein expression was induced at an OD600 of 0.6–0.8 by 0.5 mM IPTG at 18 °C for 14 h. Cells were harvested by centrifugation (8000 × g; 3 min; 4 °C) and resuspended in 300 mL lysis Buffer A (40 mM Tris-HCl, pH 7.7, 200 mM NaCl, 5% glycerol, 2 mM EDTA, 2 mM DTT, 0.1 mM phenylmethylsulfonyl fluoride (PMSF) and protease inhibitor cocktail (Biomake.cn. Inc.)) and lysed using an Avestin EmulsiFlex- C3 cell disrupter (Avestin, Inc.). The lysate was centrifuged (16,000 × g; 50 min; 4 °C), and the supernatant was precipitated with dropwise addition of 10% polyethylenimine (PEI) to a final concentration of 0.6%, and stirred at 4 °C for 30 min The pellet was collected by centrifugation (10 min; 12,000 × g) and RNAP was extracted from the pellet with 100 mL of 10 mM Tris-HCl, pH 7.7, 5% glycerol, 1 M NaCl, 1 mM DTT, and 2 mM EDTA. The RNAP was precipitated from the supernatant by addition of ammonium sulfate (final concentration; 26 g/100 ml) into the supernatant, stirred at 4 °C for 30 min, followed centrifugation at 15,000 × g for 30 min The pellet was collected, dissolved with 90 ml NTA-binding buffer (10 mM Tris-HCl, pH 7.7, 5% glycerol, 400 mM NaCl, 5 mM β-mercaptoethanol), and loaded

on to a 4-ml column packed with Ni-NTA agarose (SMART, Inc.). The bound-RNAP was sequentially washed with 100 ml NTA-binding buffer containing 0 mM, 10 mM, and 20 mM, and eluted with Ni-NTA buffer containing 300 mM imidazole. The eluted fractions were mixed with TGED buffer (10 mM Tris-HCl, pH 7.7, 5% glycerol, 2 mM DTT, 2 mM EDTA) at ratio 1:1 and loaded onto a Mono Q column (MonoQ 10/100 GL, GE healthcare Life Sciences) followed by a salt gradient of buffer A (10 mM Tris-HCl, pH 7.7, 200 mM NaCl, 5% (v/v) glycerol, 1 mM DTT, 0.1 mM EDTA) and buffer B (10 mM Tris-HCl, pH 7.7, 600 mM NaCl, 5% (v/v) glycerol, 1 mM DTT, 0.1 mM EDTA). The fractions containing target proteins were collected, concentrated to 10 mg/ml, and stored at −80 °C. The RNAP derivatives were prepared by the same procedure.

The Xp10 P7 was over-expressed in E. coli BL21(DE3) cells (Novo protein, Inc.) containing pET28a-TEV-P7. The protein expression was induced with 0.3 mM IPTG at 18 °C for 14 h when $OD_{600}$ reached to 0.6–0.8. The cell pellet was lysed in lysis buffer B (50 mM Tris-HCl, pH 7.7, 500 mM NaCl, 5% (v/v) glycerol, 5 mM β-mercaptoethanol, and protease inhibitor cocktail using a Avestin EmulsiFlex-C3 cell disrupter (Avestin, Inc.). The lysate was centrifuged ($16,000 \times g$; 50 min; 4 °C) and the supernatant was loaded on to a 2-ml column packed with Ni-NTA agarose (SMART, Inc.). The bound proteins were washed by the lysis buffer B containing 20 mM imidazole and eluted with the lysis buffer B containing 300 mM imidazole. The eluted fractions were transferred to a dialysis bag and added with TEV protease to cleave the tag while exchanging the buffer to 20 mM Tris-HCl, pH 7.7, 100 mM NaCl, 5% (v/v) glycerol, and 5 mM β-mercaptoethanol. The sample was reloaded onto the Ni-NTA column and the tag-free protein was retrieved from the flow-through fraction. The sample was diluted, loaded onto a Q HP column (HiPrep Q HP 16/10, GE healthcare Life Sciences) and eluted with a salt gradient of buffer A (20 mM Tris-HCl, pH 7.7, 0.1 M NaCl, 5% (v/v) glycerol, 1 mM DTT) and buffer B (20 mM Tris-HCl, pH 7.7, 1 M NaCl, 5% (v/v) glycerol, 1 mM DTT). The fractions containing target proteins were collected, concentrated to 5 mg/ml, and stored at −80 °C. The Xoo NusA, σ[70] and P7 derivatives were prepared by the same procedure.

**Nucleic-acid scaffolds.** Nucleic-acid scaffolds for cryo-EM study of Xoo P7-TEC and P7-NusA-TEC and for fluorescence polarization study were prepared from synthetic oligos (sequences in Fig. 1a; Supplementary Fig. 1H, 5J, and 7E) by an annealing procedure (95 °C, 5 min followed by 2 °C-step cooling to 25 °C) in annealing buffer (5 mM Tris-HCl, pH 8.0, 200 mM NaCl, and 10 mM $MgCl_2$).

**Complex reconstitution of Xoo P7-TEC and P7-NusA-TEC.** Xoo RNAP core enzyme and the nucleic-acid scaffold were incubated in a 1:1.3 molar ratio at room temperature for 15 min, 5 molar ratio of P7 (and 5 molar ratio of NusA in case of P7-NusA-TEC) was subsequently added and the mixture was further incubated overnight. The mixture was applied to a Superose 6 10/300 GL column (GE Healthcare Life Sciences) equilibrated in 10 mM HEPES, pH 7.5, 50 mM KCl, 5 mM $MgCl_2$, 3 mM DTT. Fractions containing Xoo P7-TEC or P7-NusA-TEC were collected and concentrated to 9.5 mg/ml and 12 mg/ml, respectively.

**Cryo-EM structure determination of Xoo P7-TEC and P7-NusA-TEC.** The Xoo P7-TEC and P7-NusA-TEC samples were freshly prepared as described above and mixed with CHAPSO (Hampton Research, Inc.) to a final concentration 8 mM prior to grid preparation. About 4 µL of the complex sample was applied onto the glow-discharged C-flat CF-1.2/1.3 400 mesh holey carbon grids (Electron Microscopy Sciences) and the grid was plunge-frozen in liquid ethane using a Vitrobot Mark IV (FEI) with 95% chamber humidity at 10 °C.

The data were collected on a 300 keV Titan Krios (FEI) equipped with a K2 Summit direct electron detector (Gatan). A total of 2271 images of P7-TEC and 3702 images of P7-NusA-TEC were recorded using the Serial EM[67] in super-resolution counting mode with a pixel size of 0.507 Å, and a dose rate of 6.7 electrons/pixel/s. Movies were recorded at 250 ms/frame for 8 s (32 frames total) and defocus range was varied between 2.0 µm and 2.6 µm. Frames in individual movies were aligned using MotionCor2[68], and contrast-transfer-function estimations were performed using CTFFIND4[69]. About 1000 particles were picked and subjected to 2D classification in RELION 3.0[70]. The resulting distinct two-dimensional classes were served as templates and a total of 444,591 particles for P7-TEC and 390,526 particles for P7-NusA-TEC were picked out. The resulting particles were manually inspected and subjected to 2D classification in RELION 3.0 by specifying 100 classes[70]. Poorly populated classes were removed. We used a 60-Å low-pass-filtered map calculated from structure of E. coli RNAP core enzyme[27] (PDB: 6ALH) as the starting reference model for 3D classification. Classes were combined and the particle numbers used for constructing the final cryo-EM map for P7-TEC and P7-NusA-TEC are 319,613, and 204,450, respectively. The final maps were obtained through 3D auto-refinement, CTF-refinement, Bayesian polishing, and post-processing in RELION 3.0 (Supplementary Fig. 3). Gold-standard Fourier-shell-correlation analysis (FSC)[71] indicated a mean map resolution of 3.95 Å and 3.41 Å for Xoo P7-TEC and P7-NusA-TEC, respectively.

The RNAP from the cryo-EM structure of E. coli RNAP TEC[27](PDB: 6ALH), P7 from NMR structure of P7-β'NTH (PDB: 2MC6), and NusA from the cryo-EM structure of hisPEC-NusA (PDB: 6FLQ) were manually fit into the cryo-EM density map using Chimera[72]. Rigid body and real-space refinement was performed in Coot[73] and Phenix[74].

**In vitro transcription assay.** Promoters used for in vitro transcription assays were prepared by PCR using primers (forward primer: 5′-TGGCGGAGCTTCCGGTC GGCTTCCTCCA-3′; reverse primer: 5′-TGATCCCCGAGGAGAAGCAGAGGT ACC-3′) and pARTaq vectors as templates.

To study the concentration-dependent transcription inhibition by P7, DNA templates were amplified from pARTaq-100-tR2, pARTaq-N25(+ Ext-10), pARTaq-N25(Anti35 + Ext-10) using primers shown in the Supplementary data 1. The reactions were performed in transcription buffer (40 mM Tris-HCl, pH 8.0, 75 mM NaCl, 5 mM $MgCl_2$, 12.5%Glycerol, 2.5 mM DTT, and 50 µg/ml BSA). Reaction mixture (20 µL) containing RNAP holoenzyme (50 nM) and P7 (25–400 nM) were incubated for 5 min at 37 °C, then 1 µL promoter DNA (1 µM) was added and incubation for 10 min at 37 °C for open complex formation. RNA synthesis was started by the addition of 1.2 µL NTP mixture (2 mM ATP, 2 mM CTP, 2 mM GTP, and 2 mM [α-$^{32}$P]UTP (0.036 Bq/fmol)) for 15 min at 37 °C. The reactions were terminated by adding 10 µL loading buffer (8 M urea, 20 mM EDTA, 0.025% xylene cyanol, and 0.025% bromophenol blue), boiled for 2 min, and cooled down in ice for 5 min The RNA transcripts were separated by 15% urea-polyacrylamide slab gels (19:1 acrylamide/bisacrylamide) in 90 mM Tris-borate (pH 8.0) and 0.2 mM EDTA and analyzed by storage-phosphor scanning (Typhoon; GE Healthcare, Inc.). To study the transcription inhibition effect of P7 or RNAP derivatives, the in vitro transcription assay was performed as above except that a final concentration of 200 nM P7 or P7 derivates were used.

To study the concentration-dependent antitermination by P7, DNA templates containing the bacteriophage N25 promoter fused to tR2 terminator were prepared by PCR using pARTaq-100-tR2. Reaction mixture (20 µL) in transcription buffer containing RNAP holoenzyme (50 nM) and promoter DNA fragment (50 nM) were incubated for 15 min at 37 °C for open complex formation. The RNA synthesis was initiated by addition of 1 µL limited NTP mixture (100 µM ATP, 100 µM GTP, and 100 µM [α-$^{32}$P]UTP (0.73 Bq/fmol)) for 5 min at 37 °C to obtain TECs stalled at the template position G + 29 and then 1 µL P7 (1–8 µM; stock concentration) was added, RNA elongation was resumed by the addition of 1 µL of the full-set NTP mixture (2 mM ATP, 2 mM CTP, 2 mM GTP, and 2 mM UTP; stock concentration). To only allows single-round transcription, heparin was added to 0.015 mg/mL together with the full-set NTP mixture. The reactions were terminated by adding 10 µL loading buffer (8 M urea, 20 mM EDTA, 0.025% xylene cyanol, and 0.025% bromophenol blue), boiled for 5 min, and cooled down in ice for 5 min The RNA transcripts were separated by 15% urea-polyacrylamide slab gels (19:1 acrylamide/bisacrylamide) in 90 mM Tris-borate, pH 8.0 and 0.2 mM EDTA and analyzed by storage-phosphor scanning (Typhoon; GE Healthcare, Inc.). To study the antitermination effect of P7 or RNAP derivatives, the in vitro transcription assay was performed as above except that a final concentration of 200 nM P7 or P7 derivates were used.

To study the effect of P7 on transcription pause, DNA templates containing the N25 promoter fused to his pause and tR2 terminator or N25 promoter fused to his (-hairpin) pause and tR2 (-hairpin) terminator were prepared by PCR using pARTaq-N25-his-tR2 and pARTaq-N25-his(-HP)-tR2(-HP) as template. The reactions mixture (10 µL) in transcription buffer containing RNAP holoenzyme (100 nM) or promoter DNA fragment (100 nM) were incubated for 10 min at 25 °C for open complex formation. The reactions were initiated by addition of 1.2 µL limited-set NTP mixture (100 µM ATP, 100 µM GTP, and 20 µM [α-$^{32}$P]UTP (3.6 Bq/fmol); stock concentration) for 10 min at 25 °C to obtain TECs stalled at the template position G + 29. Subsequently, 1 µL P7 (4 µM; stock concentration) and 1 µL Heparin (0.3 mg/mL; stock concentration) were added and incubated for 3 min at 25 °C. RNA extensions were resumed by addition of 1 µL of the full-set NTP mixture (100 µM ATP, CTP, GTP, and UTP each; stock concentration). The reactions were terminated at specified time points, and the RNA transcripts were separated and analyzed as above.

**Fluorescence labeling of P7.** P7 (R73C) was labeled with fluorescein at residues C73. The labeling reaction mixture (2 mL) containing P7 (0.07 mM) and Fluorescein-5-Maleimide (0.7 mM Thermo Scientific, Inc.) in 10 mM Tris-HCl, pH 7.7, 100 mM NaCl, 1% glycerol was incubated for 2 h at room temperature, and. The reaction was terminated by addition of 2 µL DTT (1 M; stock concentration), and loaded onto a 5-mL PD-10 desalting column (Biorad, Inc.). The fractions containing labeled protein was pooled and concentrated to 3 mg/mL.

**Fluorescence polarization assay.** To measure the binding affinity of P7 to RNAP or TEC, the fluorescein-labeled P7 (R73C) (5 nM) was incubated with Xoo RNAP core enzyme, RNAP holoenzyme, or TEC (5, 10, 20, 40, 80, 160, and 320 nM) in 100 µL FP-A buffer (10 mM Tris-HCl, pH 7.7, 100 mM NaCl, 1 mM DTT, 1% glycerol, and 0.025% Tween-20) in a 96-well plate (Corning, Inc) for 5 min at room temperature. The fluorescence polarization (FP) signals were measured using a plate reader (SPARK, TECAN Inc.) equipped with excitation filter of 485/20 nm and emission filter of 520/20 nm. The data were plotted in SigmaPlot (Systat software, Inc.) and the dissociation constant Kd were estimated by fitting the data to the following equation,

$$F = B[S]/(Kd + [S]) + F0 \qquad (1)$$

Where F is the FP signal at a given concentration of RNAP or TEC, F0 is the FP

signal in the absence of RNAP or TEC, [S] is the concentration of RNAP or TEC, and B is an unconstrained constant.

To measure the binding affinity of P7 to RNAP holoenzyme in the presence of RNAP inhibitors, the fluorescein-labeled P7 (R73C) (5 nM) was incubated with *Xoo* RNAP core enzyme (5, 10, 20, 40, 80, 160, and 320 nM) in the presence of lipiarmycin (32 μM) or rifampicin (1 μM) in 100 μL FP-A buffer in a 96-well plate (Corning, Inc) for 5 min at room temperature. The FP signals were measured and data were analyzed as above.

To measure the binding affinity of elongation nucleic-acid scaffolds to *Xoo* RNAP core enzyme, the Cy5-labeled elongation nucleic-acid scaffold (8 nM) was incubated with *Xoo* RNAP core enzyme (8, 16, 32, 64, 128, and 256 nM) in the presence or absence of P7 (2 μM) in 100 μL FP-B buffer (10 mM Tris-HCl, pH 7.7, 300 mM NaCl, 1 mM DTT, 1% glycerol, and 0.025% Tween-20) in a 96-well plate (Corning, Inc) for 5 min at room temperature. The FP signals were measured with the excitation filter of 635/35 nm and the emission filter of 665/8 nm. The data were plotted and $Kd$ values were fitted as above.

To measure the binding affinity of promoter DNA to *Xoo* RNAP holoenzyme, the Cy5-labeled promoter DNA (8 nM) mixed with 0.015 mg/mL heparin was incubated with *Xoo* RNAP holoenzyme (8, 16, 32, 64, 128, 256, 512, and 1024 nM) in the presence or absence of P7 (8 μM) in 100 μL the modified FP-B buffer in a 96-well plate (Corning, Inc) for 5 min at room temperature. The FP signals were measured and data were analyzed as above.

**Stopped-flow assay.** The stopped-flow assay was performed essentially as in[40]. Briefly, to monitor the promoter melting by *Xoo* RNAP holoenzymes, 60 μL *Xoo* σ70-RNAP holoenzyme (400 nM) and 60 μL Cy3-λ $P_R$ promoter DNA (10 nM) in 10 mM Tris-HCl, pH 7.7, 300 mM NaCl, 10 mM $MgCl_2$, 1 mM DTT were rapidly mixed and the change of Cy3 fluorescence was monitored in real time by a stopped-flow instrument (SX20, Applied Photophysics Ltd, UK) equipped with an excitation filter (515/9.3 nm) and a long-pass emission filter (570 nm). Lipiarmycin (120 μM; final concentration) or P7 (8 μM; final concentration) were preincubated with *Xoo* σ70-RNAP holoenzyme when specified.

**Circular dichroism spectroscopy.** Circular dichroism experiments were performed on a Chirascan spectrometer (Applied Photophysics Ltd, UK) using 0.1–0.2 mg/ml proteins in 2 mM $K_2HPO_4/KH_2PO_4$, pH 7.4 and 20 mM $(NH_4)_2SO_4$. The ellipticity values (millidegree; mdeg) were recorded at 23 °C between 240 and 185 nm with a 1-nm step size and an integration time of 0.5 s. A total of three accumulations were averaged and the buffer spectrum, which was obtained under identical conditions, was subtracted. The observed spectra were converted to mean residue ellipticity (mdg cm$^2$/dmol) by using Pro-Data viewer software.

**Quantification and statistical analysis.** All biochemical assays were performed at least three times independently. Data were analyzed with SigmaPlot 10.0 (Systat Software Inc.).

**Reporting summary.** Further information on research design is available in the Nature Research Reporting Summary linked to this article.

## Data availability

Cryo-EM data have been deposited in the RCSB Protein Data Bank (www.pdb.org) and in the Electron Microscopy Data Bank (www.emdatabank.org). The accession number for the coordinate and cryo-EM map for Xoo P7-TEC are 6J9F and EMD-9786, respectively. The accession number for the coordinate and cryo-EM map for Xoo P7-NusA-TEC are PDB: 6J9E and EMD-9785, respectively. The source data underlying Figs. 2c, 2f, 3h–k, 5f–i, and Supplementary Figs. 1e–g, 5f, g, 5k, and 7a are provided as a Source Data file. Other data are available from the corresponding authors upon reasonable request.

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

## Acknowledgements

The work was supported by the Strategic Priority Research Program of the Chinese Academy of Sciences (XDB29020000), the National Natural Science Foundation of China (31822001), and the Leading Science Key Research Program of CAS (QYZDB-SSW-SMC005). We thank Dr. Richard Ebright (Rutgers University) for the generous gift of pARTaq-N25–100-tR2 plasmid, Dr. Zhaocai Zhou for generous gift of pET28a-TEV plasmid, Dr. Zuhua He (SIPPT, CAS) for generous gift of *Xanthomonas oryzae* pv. *oryzae*, Dr. Shenghai Chang at the cryo-EM center of Zhejiang University for assistance of grid preparation and data collection, Xiang Zhang (Shanghai YueXin Life-Science Information Technology Co.) and Dr. Sheng Wang (King Abdullah University of Science & Technology) for improving resolution during cryo-EM data processing. We thank the state key laboratory of bioorganic and natural products chemistry at Shanghai institute of organic chemistry at CAS for sharing the stopped-flow fluorescence spectrometer.

## Author contributions

L.Y. solved the cryo-EM structures and performed biochemical experiments. J.S. collected the cryo-EM data. L.S. and C.F. assisted in structure determination. L.L. performed the stopped-flow assay. C.Y. and W.C. assisted in protein purification and in vitro transcription experiments. Y.F. and Y.Z. designed experiments, analyzed data, and wrote the manuscript.

## Additional information

**Competing interests:** The authors declare no competing interests.

