## [Peer Review File · Nature Communications]

Reviewers' comments:

Reviewer #1 (Remarks to the Author):

This is an important paper that provides high quality structural information on a bifunctional phage-encoded transcription factor and gives structural background to the previously known effects of XP10 p7 on transcription initiation. In so doing it also provides insights into mechanisms of transcription antitermination and initiation. It deserves a prominent publication.

The language suffers at times and will require careful proofing.

On page 5 it is stated that p7 is the only known bifunctional transcription factor. This is certainly not true" the Q protein of phage lambda is another, as is gp39 encoded by some Thermus phages.

The experiment in Fig. 3H and the corresponding text do not add much to what was previously known from biochemical studies, it is worth omitting from the paper.

Reviewer #2 (Remarks to the Author):

Bacteriophages use numerous ways to modify transcription system in their hosts to express their genes and shutdown host gene expression. In this study, authors used the cryo-EM structure determination and biochemical approaches to reveal the interaction between RNA polymerase from *Xanthomonas oryzae* pv. *Oryzae* (rice bacterial pathogen) and P7 protein from bacteriophage Xp10. High-resolution structures of RNAP and P7 complex in two distinct forms (elongation complex with or without elongation factor NusA) were determined that reveal the P7 binding site on RNAP and how it inhibits the transcription termination by preventing the termination hairpin formation as well as inhibits the transcription initiation by preventing the binding of a domain of promoter recognition sigma factor. Authors presented structures nicely to explain and/or predict the mechanisms of P7-dependent RNAP modification and predictions were confirmed by structure-based biochemical experiments. This study also provides a structural insight into the transcription termination of bacterial RNAP, which has not been studied due to its instability. A manuscript is well-written and figures are designed in high-quality. I recommend this manuscript to be published in Nature communication with some modifications.

Major critique:

Authors used the RNAP conformation, closed RNAP clamp, observed in the cryo-EM structure of EC with P7 to proposed the mechanism of transcription inhibition by P7. The closed RNAP clamp has been observed in the most of transcription elongation complex structures containing the DNA/RNA scaffold, which was used for preparing the EC in this study. Therefore, the closed clamp conformation found in this study is due to the RNAP forming the EC but not the binding P7 to RNAP. Authors have to determine the structure of RNAP-P7 complex without EC scaffold, or determine the clamp conformation by the FRET study reported by the Ebright's group. Nonetheless, following biochemical experiments (Figs. 5E-I) are strong enough to propose the transcription inhibition mechanism. Therefore, I ask authors to modify the first two paragraphs of the section (P7 inhibits transcription initiation by jamming the RNAP clamp) to tone down how the structure was used for predicting the mechanism.

Minor comments:

Authors describe more about *Xanthomonas oryzae* pv. *Oryzae*, such as it belongs to gamma proteobacteria so RNAP from this bacterium is similar to *E. coli*, etc...

Fig. 6: RNAP from *Xoo* is closely related to *E. coli*. Why authors used the Taq RPo instead of the *E. coli* RPo?

SFigs. 2H and 2K: None of 2D classes presented here doesn't look like *E. coli* RNAP. And these 2D images do not show structural detail although the final 3D structures were determined in good

resolutions. Authors make sure that they are posting the 2D classes used for this study.

Reviewed by Katsuhiko Murakami, Penn State University

Reviewer #3 (Remarks to the Author):

The manuscript describes two structures of the transcription elongation complex (TEC) of bacterial RNA polymerase containing a phage-encoded antiterminator protein (P7 of phage Xp10). Based on these structures and biochemical evidence a mechanism of transcription antitermination is proposed, which is based on the inhibition of RNA folding in the RNA exit channel of RNAP. Interestingly, P7 seems to redirect RNA through an alternative exit path. Furthermore, the data suggest a detailed mechanism of transcription inhibition by P7 during promoter recognition. Given that this is the first high resolution structure of a complete TEC bound to an antitermination factor (and the first structure of RNAP from *X. oryzae*, which is an important rice pathogen), the manuscript may be of sufficient interest to be published in Nature Communications. Some specific comments that should be addressed before publication are listed below.

- 1) Some of the P7 mutations tested here could affect not only a specific interface at its binding site on RNAP, but also overall P7 structure (especially when a small hydrophilic or charged residue is replaced with tryptophan). Have the effects of these substitutions on p7 folding been tested?
- 2) Is the observed interaction of P7 with the flap domain important for its binding to RNAP and for its effects on transcription initiation and termination? As was shown previously, it may be important for inhibition of initiation (Liu et al., NAR 2014), but not for antitermination (Esyunina et al., 2015) – now, with the detailed structure available, this interface could be probed more specifically.
- 3) It seems that the antitermination activity of P7 is weaker in the absence of the C-terminal helix of omega (Suppl. Figure 5F). There are two RNA bands in the terminator area, only the upper one is affected by P7 (not only here but also on other Figures), and the intensity of this band is significantly higher in the case of mutant RNAP – was only this upper band counted?
- 4) The authors ignore an essential publication about p7 in the Introduction and Discussion – the 2015 paper by Zenkin et al (PMID: 26038312). It was shown there that p7 promotes forward translocation by RNAP, which may contribute to its antipausing and antitermination effects. It must be included in the manuscript, together with discussion how these biochemical observations can be reconciled with the structural models described here.
- 5) In their comparison of known phage-encoded antiterminators, the authors missed another important protein, gp39 from phage P23-45 targeting *Thermus thermophilus* RNAP. The effects of this protein on different steps of transcription have been studied in considerable detail (Berdygulova et al., 2011, PMID 21050864; Berdygulova et al., 2012, PMID 22238378; Tagami et al., 2014, PMID 24589779; Ooi et al., 2017, PMID 29165680). It was shown that, similarly to p7, gp39 inhibits transcription initiation but stimulates transcription elongation by suppressing transcriptional pausing. It also interacts with the flap domain (which is its main binding site on RNAP) and inhibits interactions of region sigma4 with the -35 promoter element. Furthermore, it may likely change the position of the flap domain thus modulating pausing. However, in contrast to p7 its activity is not stimulated by NusA, likely because the two proteins bind to overlapping sites on RNAP and thus compete with each other for binding. All relevant papers should be properly cited here.
- 6) Structures of intermediate complexes of bacterial RNAP during promoter melting have been published recently (Boyaci et al., 2019, PMID 30626968); these structures may be relevant to the proposed mechanism of inhibition of transcription initiation by p7 through prevention of clamp opening (Figure 7B), and should be discussed here.
- 7) Furthermore, regulation of clamp opening (during promoter opening and pausing) may depend on ppGpp and the omega subunit, which constitutes a part of its binding interface (e.g., Duchi et al., 2018, PMID 29878276) – could any interplay between p7, omega, ppGpp, clamp opening and p7 be expected based on the present structures?
- 8) The effects of some P7 mutations on its activity have been characterized previously (Liu et al., 2014 NAR, PMID 24482445; Liu et al., 2014 Bacteriophage, PMID 24701369) and should be mentioned here.

- 9) Page 9, middle paragraph, second sentence – the effects of P7 on elemental (hairpin-less) and U-tract-induced pauses have been studied previously (Esyunina et al., 2015), which should be mentioned here or in Discussion.
- 10) P7 does not significantly affect pausing on terminator-derived oligoU sequences (Esyunina et al., 2015 and this manuscript) but it does suppress pausing on oligoU-tract in the presence of NusA (Esyunina et al., 2015) – what is the explanation from the structural point of view?
- 11) Given that the affinity of p7 to the TEC is in the order of 10 nM (panels E-G in Supplementary Figure 1), why much more p7 is needed to see its effects on transcription (panels A and B in the same figure)? RNAP-TEC concentration in panel A is 50 nM, so complete TEC saturation with p7 should occur at 50 nM, not 200 nM seen here.
- 12) It is interesting that p7 is expected to prevent transcription initiation by cellular RNAP from all types of promoters but at the same time to act as an antiterminator, suggesting that some RNAP should still be engaged in RNA synthesis – how these two activities can be reconciled in an integrated model of transcription regulation?
- 13) Changes in the position and conformation of NusA in comparison with other published structures of the E. coli RNAP TEC bound to NusA (paused complex with and without hairpin, complex with lambda N) should be discussed in more detail here (preferably with a structural illustration).
- 14) Page 13, last sentence – The structure presented here does not demonstrate that the hairpin invasion is the obligatory step of termination, need to re-write.
- 15) The manuscript title is misleading because no structural information for an intrinsic termination complex is presented here.
- 16) Page 5, line 2 – Ref 24 is wrong here, the paper by Berdygulova et al is about gp39, not p7.
- 17) Page 14, line 6 - "...and thus prevent hairpin formation".
- 18) Page 16, fourth line from the bottom – "at ratio 1:1".
- 19) Page 18, fifth line from the bottom – 1 uM to 8 uM of P7, is it final or initial concentration before p7 addition?
- 20) Page 19, second paragraph, fifth line from the bottom – is 4 uM starting or final P7 concentration?
- 21) Ref 27 is identical to ref 23.
- 22) Supplementary Figure 1 legend, last sentence – "used in (G)" seemingly.

Response to reviewers' comments:

Reviewer #1 (Remarks to the Author):

This is an important paper that provides high quality structural information on a bifunctional phage-encoded transcription factor and gives structural background to the previously known effects of XP10 p7 on transcription initiation. In so doing it also provides insights into mechanisms of transcription antitermination and initiation. It deserves a prominent publication.

The language suffers at times and will require careful proofing.

On page 5 it is stated that p7 is the only known bifunctional transcription factor. This is certainly not true" the Q protein of phage lambda is another, as is gp39 encoded by some Thermus phages.

The experiment in Fig. 3H and the corresponding text do not add much to what was previously known from biochemical studies, it is worth omitting from the paper.

Reply:

Thanks for the comment. We have deleted the sentence “P7 is the only reported phage protein with dual regulatory functions on host RNA polymerase” on page 5. Moreover, we have added the following discussion on page 14 to the revised manuscript.

“The gp39 also resides at outer rim of the RNA exit channel by interacting with both σ^A_4 and RNAP- β flap domain. Although the structure of gp39-engaged *T. thermophilus* RNAP holoenzyme couldn't explain how gp39 inhibits transcription termination, subtle movement of gp39 induced by dissociation of σ^A would possibly bring gp39 closer to the RNA exit channel and consequently interfere with RNA hairpin formation in the channel^{24,58,59,60}. Our structure provides the first structural explanation how inhibition of hairpin formation is achieved by the phage regulatory factor P7 (Fig. 7a). As all of the reported transcription antitermination factors (λ N, Q, and gp39) locate at sites near the RNA exit channel, they probably prevent transcription termination through a unified mechanism.”

The Fig. 3h and Fig. 3i are closely related; and the promoter DNA sequences are exactly the same except that the hairpin-encoding sequences were deleted in Fig. 3h. We think it is better to include in the manuscript for a direct comparison of P7's effects on hairpin-containing and hairpin-less DNA.

Reviewer #2 (Remarks to the Author):

Bacteriophages use numerous ways to modify transcription system in their hosts to express their genes and shutdown host gene expression. In this study, authors used the cryo-EM structure determination and biochemical approaches to reveal the interaction between RNA polymerase from *Xanthomonas oryzae* pv. *Oryzae* (rice bacterial pathogen) and P7 protein from bacteriophage Xp10. High-resolution structures of RNAP and P7 complex in two distinct forms (elongation complex with or without elongation factor NusA) were determined that reveal the P7 binding site on RNAP and how it inhibits the transcription termination by

preventing the termination hairpin formation as well as inhibits the transcription initiation by preventing the binding of a domain of promoter recognition sigma factor. Authors presented structures nicely to explain and/or predict the mechanisms of P7-dependent RNAP modification and predictions were confirmed by structure-based biochemical experiments. This study also provides a structural insight into the transcription termination of bacterial RNAP, which has not been studied due to its instability. A manuscript is well-written and figures are designed in high-quality. I recommend this manuscript to be published in Nature communication with some modifications.

Reviewer #2, Major critique;

Authors used the RNAP conformation, closed RNAP clamp, observed in the cryo-EM structure of EC with P7 to proposed the mechanism of transcription inhibition by P7. The closed RNAP clamp has been observed in the most of transcription elongation complex structures containing the DNA/RNA scaffold, which was used for preparing the EC in this study. Therefore, the closed clamp conformation found in this study is due to the RNAP forming the EC but not the binding P7 to RNAP. Authors have to determine the structure of RNAP-P7 complex without EC scaffold, or determine the clamp conformation by the FRET study reported by the Ebright's group. Nonetheless, following biochemical experiments (Figs. 5E-I) are strong enough to propose the transcription inhibition mechanism. Therefore, I ask authors to modify the first two paragraphs of the section (P7 inhibits transcription initiation by jamming the RNAP clamp) to tone down how the structure was used for predicting the mechanism.

Reply:

Thanks for the suggestion. We have added the following sentence in the revised manuscript on p11.

“The RNAP clamp of P7-TEC-NusA structure adopts the closed conformation (Fig. 5a and Supplementary Fig. 4f), which has been observed in *E. coli* TEC structure containing similar nucleic-acid scaffold²⁸”

Reviewer #2, minor comment 1:

Authors describe more about *Xanthomonas oryzae* pv. *Oryzae*, such as it belongs to gamma proteobacteria so RNAP from this bacterium is similar to *E. coli*, etc...

Reply:

Thanks for the suggestion. We have added the sentence into the revised manuscript.

“The *Xoo* belongs to γ proteobacteria and thereby the RNAP from this bacterium is similar to that of *E. coli*.”

Reviewer #2, minor comment 2:

Fig. 6: RNAP from *Xoo* is closely related to *E. coli*. Why authors used the *Taq* RPo instead of the *E. coli* RPo?

Reply:

We have replaced the *T. aquaticus* RPo with *E. coli* RPo for comparison and also replace the related citation.

Reviewer #2, minor comment 3:

SFigs. 2H and 2K: None of 2D classes presented here doesn't look like *E. coli* RNAP. And

these 2D images do not show structural detail although the final 3D structures were determined in good resolutions. Authors make sure that they are posting the 2D classes used for this study.

Reply:

We confirm that they are the correct images.

Reviewer #3 (Remarks to the Author):

The manuscript describes two structures of the transcription elongation complex (TEC) of bacterial RNA polymerase containing a phage-encoded antiterminator protein (P7 of phage Xp10). Based on these structures and biochemical evidence a mechanism of transcription antitermination is proposed, which is based on the inhibition of RNA folding in the RNA exit channel of RNAP. Interestingly, P7 seems to redirect RNA through an alternative exit path. Furthermore, the data suggest a detailed mechanism of transcription inhibition by P7 during promoter recognition. Given that this is the first high resolution structure of a complete TEC bound to an antitermination factor (and the first structure of RNAP from *X. oryzae*, which is an important rice pathogen), the manuscript may be of sufficient interest to be published in Nature Communications. Some specific comments that should be addressed before publication are listed below.

Reviewer #3, comment 1;

Some of the P7 mutations tested here could affect not only a specific interface at its binding site on RNAP, but also overall P7 structure (especially when a small hydrophilic or charged residue is replaced with tryptophan). Have the effects of these substitutions on p7 folding been tested?

Reply:

We have not specifically tested the folding of P7 derivatives, but all the derivatives were obtained in soluble fraction when over-expressed in *E. coli*, indicating no major folding issue occurred.

Reviewer #3, comment 2;

Is the observed interaction of P7 with the flap domain important for its binding to RNAP and for its effects on transcription initiation and termination? As was shown previously, it may be important for inhibition of initiation (Liu et al., NAR 2014), but not for antitermination (Esyunina et al., 2015) – now, with the detailed structure available, this interface could be probed more specifically.

Reply:

Our structure shows that P7 approaches to but doesn't make interaction with the flap domain of RNAP- β subunit (β flap) in both P7-TEC and P7-TEC-NusA structures (Fig. 2a and Supplementary Fig. 4d). P7 has a relatively large hydrophobic interface for interactions with the β 'NTH. It might interact with the hydrophobic tip helix of β flap when only the two components were present in the BTH assay in Liu et al., NAR 2014. Moreover, the proposed interface residue (R60 of P7) in that study is at the opposite site of RNAP- β flap in our structure, where it makes interaction with RNAP- β 'dock. Therefore, the impaired ability for transcription inhibition of P7(R60A) derivative actually resulted from the impaired interaction between P7 and RNAP- β 'dock. The absence of interactions between P7 and β flap also explains that the β flap is dispensable for P7-mediated transcription antitermination in

Esyunina et al., 2015.

Reviewer #3, comment 3;

It seems that the antitermination activity of P7 is weaker in the absence of the C-terminal helix of omega (Suppl. Figure 5F). There are two RNA bands in the terminator area, only the upper one is affected by P7 (not only here but also on other Figures), and the intensity of this band is significantly higher in the case of mutant RNAP – was only this upper band counted?

Reply:

Yes, only the intensity of the upper band was quantitated throughout the manuscript. As pointed out by the reviewer, only the upper band was affected by P7 and should be terminated transcripts.

Reviewer #3, comment 4;

The authors ignore an essential publication about p7 in the Introduction and Discussion – the 2015 paper by Zenkin et al (PMID: 26038312). It was shown there that p7 promotes forward translocation by RNAP, which may contribute to its antipausing and antitermination effects. It must be included in the manuscript, together with discussion how these biochemical observations can be reconciled with the structural models described here.

Reply:

We have added the citation and the following discussions in the revised manuscript, on P9

“The much narrower alternative RNA gate in the P7-TEC should interact with the ssRNA more tightly than the regular one and thus might account for the effect of promoting forward translocation by P7”

on P12,

“The P7-induced closed conformation of RNAP-clamp also explains the effect of promoting forward translocation by P7 during transcription elongation ²⁴”

Reviewer #3, comment 5;

In their comparison of known phage-encoded antiterminators, the authors missed another important protein, gp39 from phage P23-45 targeting *Thermus thermophilus* RNAP. The effects of this protein on different steps of transcription have been studied in considerable detail (Berdygulova et al., 2011, PMID 21050864; Berdygulova et al., 2012, PMID 22238378; Tagami et al., 2014, PMID 24589779; Ooi et al., 2017, PMID 29165680). It was shown that, similarly to p7, gp39 inhibits transcription initiation but stimulates transcription elongation by suppressing transcriptional pausing. It also interacts with the flap domain (which is its main binding site on RNAP) and inhibits interactions of region sigma4 with the -35 promoter element. Furthermore, it may likely change the position of the flap domain thus modulating pausing. However, in contrast to p7 its activity is not stimulated by NusA, likely because the two proteins bind to overlapping sites on RNAP and thus compete with each other for binding. All relevant papers should be properly cited here.

Reply:

Thanks for pointing out the phage protein gp39 we have missed. During the review process, a report of a high-resolution structure of λ N-TAC was online. Therefore, we have added the citations and corresponding texts for gp39 and also for λ N in the discussion section (p14),

“The recent high-resolution cryo-EM structure of λ N-TAC reveals that the C-terminal loop of λ N penetrates the RNA exit channel and probably compete with RNA hairpin formation in the channel³⁵. Extensive biochemical evidence suggests that Q also locates near the RNA exit channel to prevent transcription termination, but the structural basis is unknown^{57,58,59}. The gp39 also resides at outer rim of the RNA exit channel by interacting with both σ^A_4 and RNAP- β flap domain. Although the structure of gp39-engaged *T. thermophilus* RNAP holoenzyme couldn't explain how gp39 inhibits transcription termination, subtle movement of gp39 induced by dissociation of σ^A would possibly bring gp39 closer to the RNA exit channel and consequently interfere RNA hairpin formation in the channel^{24,58,59,60}. Our structure provides the first structural explanation how inhibition of hairpin formation is achieved by the phage regulatory factor P7 (Fig. 7a). As all the reported transcription antitermination factors (λ N, Q, and gp39) locate at sites near the RNA exit channel, they probably prevent transcription termination by the mechanism similar to P7.”

Reviewer #3, comment 6;

Structures of intermediate complexes of bacterial RNAP during promoter melting have been published recently (Boyaci et al., 2019, PMID 30626968); these structures may be relevant to the proposed mechanism of inhibition of transcription initiation by p7 through prevention of clamp opening (Figure 7b), and should be discussed here.

Reply:

The structures have already been cited in the manuscript. We have also modified the sentence in Page 15,

“It is proposed that the RNAP clamp remains predominately open for scanning in the genome, closes transiently to nucleate the promoter melting at the -10 element³⁹, reopens twice to sequentially clear two physical barrier (the β -lobe-gate barrier and FL2-Sw2 barrier) for loading promoter DNA into the RNAP main cleft¹³, and finally form a catalytic-competent RPO^{38,39,42}.”

Reviewer #3, comment 7;

Furthermore, regulation of clamp opening (during promoter opening and pausing) may depend on ppGpp and the omega subunit, which constitutes a part of its binding interface (e.g., Duchi et al., 2018, PMID 29878276) – could any interplay between p7, omega, ppGpp, clamp opening and p7 be expected based on the present structures?

Reply:

The available crystal structure of *Ec* RNAP-ppGpp and the data from single-molecular FRET experiments suggest that ppGpp stabilizes RNAP in a partly-closed-clamp state, which might be compatible with P7 binding. Since both P7 and ppGpp act on restraining intrinsic flexibility of RNAP, they might have additive effect on inhibiting of transcription inhibition.

Reviewer #3, comment 8;

The effects of some P7 mutations on its activity have been characterized previously (Liu et al., 2014 NAR, PMID 24482445; Liu et al., 2014 Bacteriophage, PMID 24701369) and should be mentioned here.

Reply:

The work of Liu et al., 2014 NAR, PMID 24482445 has already been mentioned in the manuscript. We have added the second citation (Liu et al., 2014 Bacteriophage, PMID

24701369) in the revised manuscript.

Reviewer #3, comment 9;

Page 9, middle paragraph, second sentence – the effects of P7 on elemental (hairpin-less) and U-tract-induced pauses have been studied previously (Esyunina et al., 2015), which should be mentioned here or in Discussion.

Reply:

We have added the following sentence in the paragraph, “consistent with the previous finding that P7 itself has little effect on poly-U pause and further supporting the hypothesis (Fig. 3j-k and Supplementary Fig. 5j)²⁵”

Reviewer #3, comment 10;

P7 does not significantly affect pausing on terminator-derived oligoU sequences (Esyunina et al., 2015 and this manuscript) but it does suppress pausing on oligoU-tract in the presence of NusA (Esyunina et al., 2015) – what is the explanation from the structural point of view?

Reply:

Since the characteristic of poly-U pause remains elusive, it is difficult to explain why P7 and NusA have a synergistic effect on inhibiting poly-U pause.

Reviewer #3, comment 11;

Given that the affinity of p7 to the TEC is in the order of 10 nM (panels E-G in Supplementary Figure 1), why much more p7 is needed to see its effects on transcription (panels A and B in the same figure)? RNAP-TEC concentration in panel A is 50 nM, so complete TEC saturation with p7 should occur at 50 nM, not 200 nM seen here.

Reply:

This is a very good question. We don't have answer yet. But it could be that P7 saturates RNAP at lower concentration without any competition from promoter DNA (in the case of transcription initiation) or RNA hairpin (in the case of transcription elongation) in the FP experiments than P7 does in the *in vitro* transcription assays.

Reviewer #3, comment 12;

It is interesting that p7 is expected to prevent transcription initiation by cellular RNAP from all types of promoters but at the same time to act as an antiterminator, suggesting that some RNAP should still be engaged in RNA synthesis – how these two activities can be reconciled in an integrated model of transcription regulation?

Reply:

The phage Xp10 encodes a single-subunit RNAP, which could partially account for transcription of phage late genes when the host bacterial RNAP is totally shut down by P7.

Reviewer #3, comment 13;

Changes in the position and conformation of NusA in comparison with other published structures of the E. coli RNAP TEC bound to NusA (paused complex with and without hairpin, complex with lambda N) should be discussed in more detail here (preferably with a structural illustration).

Reply:

Thanks for the suggestion. We have included NusA from the structure of λ N-antitermination

complex into the Supplementary Fig. 6b . We also added the following discussion in the revised manuscript,

“The conformation and binding location on RNAP of NusA in our structure of P7-NusA-TEC is similar to the NusA in the structure of *E. coli* NusA-PEC³³, but distinct from the NusA in the structure of λ N-mediated transcription anti-termination complex³⁷(Supplementary Fig. 6b).”

Reviewer #3, comment 14;

Page 13, last sentence – The structure presented here does not demonstrate that the hairpin invasion is the obligatory step of termination, need to re-write.

Reply:

We have modified the sentences as follows,

“Our structure supports the proposal that the hairpin invasion is the obligatory step for intrinsic termination⁹”.

Reviewer #3, comment 15;

The manuscript title is misleading because no structural information for an intrinsic termination complex is presented here.

Reply:

Thanks for comment. We like the title because our work is the first time to reveal the structural basis and mechanism of transcription antitermination by phage proteins, which specifically render bacterial RNAP to bypass intrinsic termination signals.

Reviewer #3, comment 16;

Page 5, line 2 – Ref 24 is wrong here, the paper by Berdygulova et al is about gp39, not p7.

Reply:

Thanks. The wrong citation has been replaced by Zenkin et al., 2015, PMID: 26038321.

Reviewer #3, comment 17;

Page 14, line 6 - “...and thus prevent hairpin formation”.

Reply:

Thanks. The grammatical error has been corrected.

Reviewer #3, comment 18;

Page 16, fourth line from the bottom – “at ratio 1:1”.

Reply:

Thanks. The typo has been corrected.

Reviewer #3, comment 19;

Page 18, fifth line from the bottom – 1 μ M to 8 μ M of P7, is it final or initial concentration before p7 addition?

Reply: It is initial (stock) concentration. We have modified the corresponding words in the method section.

Reviewer #3, comment 20;

Page 19, second paragraph, fifth line from the bottom – is 4 μ M starting or final P7 concentration?

Reply: It is initial (stock) concentration. We have modified the corresponding words in the method section.

Reviewer #3, comment 21;

Ref 27 is identical to ref 23.

Reply: Corrected.

Reviewer #3, comment 22;

Supplementary Figure 1 legend, last sentence – “used in (G)” seemingly.

Reply: Thanks. The typo has been corrected.

Reviewers' comments:

Reviewer #3 (Remarks to the Author):

The authors have adequately addressed most comments – but some issues still require clarification. All points below are based on my original review, and these changes must be introduced in the manuscript text, not only explained in the response to the reviewers.

1) Some of the P7 mutations tested in the manuscript could affect not only a specific interface at its binding site on RNAP, but also overall P7 structure (especially when a small hydrophilic or charged residue is replaced with tryptophan). Have the effects of these substitutions on p7 folding been tested?

The authors' reply is that all P7 variants were expressed well in *E. coli* and were obtained in a soluble form. However, this does not guarantee that the protein is properly folded. My especial concern is about P7 variants in which negatively charged residues, Ser, or Pro were replaced with tryptophan. If folding of these variants is not analyzed directly (e.g. by circular dichroism or other techniques), this potential problem at least should be noted in the manuscript text.

2) It seems that the antitermination activity of P7 is weaker in the absence of the C-terminal helix of omega (Suppl. Figure 5F). There are two RNA bands in the terminator area, only the upper one is affected by P7 (not only here but also in other Figures), and the intensity of this band is significantly higher in the case of mutant RNAP – was only this upper band counted?

The authors reply that indeed only the upper band was used for quantification – but this must be also explicitly explained in the manuscript text. Furthermore, deletion of the omega CTH clearly increases the termination efficiency in the presence of P7 (i.e., decreases the efficiency of p7 action) – see supplementary fig. 5f. This fact also needs to be acknowledged in the manuscript.

3) Furthermore, regulation of clamp opening (during promoter opening and pausing) may depend on ppGpp and the omega subunit, which constitutes a part of its binding interface (e.g., Duchi et al., 2018, PMID 29878276) – could any interplay between p7, omega, ppGpp, clamp opening and p7 be expected based on the present structures?

The authors' reply is that indeed such interplay may be possible during transcription initiation. And what about transcriptional pausing? Previously, ppGpp was shown to stimulate the antipausing activity of P7 (Esyunina et al., 2015) – how can this be interpreted from the structural point of view? Again, some discussion in the manuscript text is needed.

4) The effects of some P7 mutations on its activity have been characterized previously (Liu et al., 2014 NAR, PMID 24482445; Liu et al., 2014 Bacteriophage, PMID 24701369) and should be mentioned here.

It is interesting that the V51A substitution strongly impaired the antitermination activity of p7 but had only small effect on the inhibition of transcription initiation (supplementary fig. 7a) – do the authors have any explanation?

5) P7 does not significantly affect pausing on terminator-derived oligoU sequences but it does suppress pausing on oligoU-tract in the presence of NusA – what is the explanation from the structural point of view?

The authors reply that it is difficult to explain – but this must be also discussed in the manuscript text. In particular, in Fig. 3h there is no effect of p7 on oligoU-pausing, but it was observed previously in the presence of NusA (Esyunina et al., 2015). One possible explanation is that in the absence of hairpin oligoU induces RNAP backtracking, while may P7+NusA prevent backtracking and promote forward translocation.

6) It is interesting that p7 is expected to prevent transcription initiation by cellular RNAP from all types of promoters but at the same time to act as an antiterminator, suggesting that some RNAP

should still be engaged in RNA synthesis – how these two activities can be reconciled in an integrated model of transcription regulation?

The authors say in their reply that transcription of late phage genes could be performed by the phage-encoded RNAP when the host is totally shut down by P7. However, this does not explain how P7 could act as an antiterminator if the host RNAP is inhibited – it is only possible if RNAP can initiate transcription.

Reviewer #3 (Remarks to the Author):

The authors have adequately addressed most comments – but some issues still require clarification. All points below are based on my original review, and these changes must be introduced in the manuscript text, not only explained in the response to the reviewers.

Comment 1:

1) Some of the P7 mutations tested in the manuscript could affect not only a specific interface at its binding site on RNAP, but also overall P7 structure (especially when a small hydrophilic or charged residue is replaced with tryptophan). Have the effects of these substitutions on p7 folding been tested?

The authors' reply is that all P7 variants were expressed well in E. coli and were obtained in a soluble form. However, this does not guarantee that the protein is properly folded. My especial concern is about P7 variants in which negatively charged residues, Ser, or Pro were replaced with tryptophan. If folding of these variants is not analyzed directly (e.g. by circular dichroism or other techniques), this potential problem at least should be noted in the manuscript text.

Reply:

Thanks for reviewer's suggestion. We have performed CD measurements for all the P7 mutants. The results did show that two tryptophan substitutions of P7 (E49W and S46W), and unexpectedly two alanine substitutions (F50A and Y44A) affected overall P7 structure. We added the CD measurements of P7 mutants into supplementary figure 5k. The potential folding issues of the four mutants have been noted in the legend of figure 2 and supplementary figure 7 in the revised manuscript. "The asterisks indicate substitutions, which

might also affect overall structure of P7.”

Although the above mutants might also affect the overall structure of P7, the three interfaces between P7 and RNAP in the structure are still validated by other substitutions (V51A and $\Delta\beta$ 'NTH for the P7/ β 'NTH interface; D43A and E53W for the P7/ β 'dock interface; and P48W and A52E for the P7/ β CTR interface), which don't affect P7 folding.

Comment 2:

2) It seems that the antitermination activity of P7 is weaker in the absence of the C-terminal helix of omega (Suppl. Figure 5F). There are two RNA bands in the terminator area, only the upper one is affected by P7 (not only here but also in other Figures), and the intensity of this band is significantly higher in the case of mutant RNAP – was only this upper band counted?

The authors reply that indeed only the upper band was used for quantification – but this must be also explicitly explained in the manuscript text. Furthermore, deletion of the omega CTH clearly increases the termination efficiency in the presence of P7 (i.e., decreases the efficiency of p7 action) – see supplementary fig. 5f. This fact also needs to be acknowledged in the manuscript.

Reply:

Thanks for the suggestion. We have modified the text in the manuscript as follows, “Truncation of the C-terminal half-helix of RNAP- ω subunit or the C-terminal loop of P7 (the two regions might interact with each other in a structure model with extent RNAP- ω C-terminal helix; Supplementary Fig. 5e) has slightly alleviated the inhibitory activity of P7 on transcription termination (Supplementary Fig. 5f), but has no obvious effect on the action of P7 on transcription initiation, consistent with the previous study²⁵”

Moreover, we have included the following description into the legends of Figure 2C and Fig S5F. “The area of terminated transcript contains two RNA bands, of which only the upper one is affected by P7 and thereby quantitated.”

Comment 3:

3) Furthermore, regulation of clamp opening (during promoter opening and pausing) may depend on ppGpp and the omega subunit, which constitutes a part of its binding interface (e.g., Duchi et al., 2018, PMID 29878276) – could any interplay between p7, omega, ppGpp, clamp opening and p7 be expected based on the present structures?

The authors' reply is that indeed such interplay may be possible during transcription initiation. And what about transcriptional pausing? Previously, ppGpp was shown to stimulate the antipausing activity of P7 (Esyunina et al., 2015) – how can this be interpreted from the structural point of view? Again, some discussion in the manuscript text is needed.

Reply:

Thanks for the suggestion. It is difficult for us at the current stage to confidently discuss the interplay between ppGpp and P7 based on the two points,

1) The available structures of RNAP containing ppGpp are determined by X-ray

crystallography, which may introduce uncertainty of the RNAP conformation in the presence of ppGpp, because the overall conformation of RNAP largely depends on crystal packing. Furthermore, the crystal structures don't contain any nucleic-acid scaffold and thereby provide limited implication for the function of ppGpp, especially during transcription elongation.

- 2) ppGpp has two binding sites (one is on the interface of RNAP- ω and β' subunits; the other one is on the interface between DksA and RNAP- β' subunit). Previous reports suggested that ppGpp played roles in transcription pausing/ intrinsic termination either in the presence (Ferman et al., 2012, NAR, PMID: 22210857) or absence of DksA (Esyunina et al., 2015, PMID 25646468). It increases the complexity of the action of ppGpp.
- 3) Therefore, considering the fact that it is unclear of the RNAP conformation in the presence of ppGpp during transcription elongation and of the mechanism by which ppGpp affects transcription termination by itself, we are not confident to discuss the interplay between P7 and ppGpp and decide not to include our premature thought in the manuscript.

Comment 4:

The effects of some P7 mutations on its activity have been characterized previously (Liu et al., 2014 NAR, PMID 24482445; Liu et al., 2014 Bacteriophage, PMID 24701369) and should be mentioned here.

It is interesting that the V51A substitution strongly impaired the antitermination activity of p7 but had only small effect on the inhibition of transcription initiation (supplementary fig. 7a) – do the authors have any explanation?

Reply:

We have mentioned and cited the references in the manuscript in the second paragraph of page 7. Please see below

“Truncation of β' NTH ($\Delta\beta'$ NTH-RNAP) or alanine substitution of nonpolar residues of P7 (F50A and V51A) causes substantial loss of antitermination effect of P7 (Fig. 2c), consistent with previous findings and suggesting a functional relevance of the interaction^{23, 26, 28}.”

V51A is not the only substitution, which shows different effects on antitermination and inhibition of transcription initiation by P7. For example, P7 failed to inhibit transcription termination but still slightly inhibits initiation of $\Delta\beta'$ NTH-RNAP. Substitutions E53W and A52W of P7 completely abolished the antitermination effect of P7 but still retained partial inhibition on transcription initiation. It seems that transcription antitermination is broadly more sensitive to the interruption of P7-RNAP interactions.

During transcription elongation, P7 prevents transcription termination by inhibiting RNA hairpin formation. Therefore, P7 is in direct competition with RNA hairpin, disrupting interactions of P7 and RNAP would significantly increase the chance of RNA-hairpin formation. However, P7 inhibits transcription initiation through dual mechanisms--by displacing sR4 and by restraining the RNAP clamp motion; thereby, disrupting certain interactions of P7 and RNAP may only significantly affect one of the dual mechanisms.

Comment 5:

P7 does not significantly affect pausing on terminator-derived oligoU sequences but it does suppress pausing on oligoU-tract in the presence of NusA – what is the explanation from the structural point of view?

The authors reply that it is difficult to explain – but this must be also discussed in the manuscript text. In particular, in Fig. 3h there is no effect of p7 on oligoU-pausing, but it was observed previously in the presence of NusA (Esyunina et al., 2015). One possible explanation is that in the absence of hairpin oligoU induces RNAP backtracking, while may P7+NusA prevent backtracking and promote forward translocation.

Reply:

Thanks for the suggestion. We have included the following discussion in the manuscript. “The interactions of NusA and P7 would further stabilize closed conformation of RNAP induced by P7 (discussed below) and consequently prevent backtracking, probably accounting for reduced pausing at poly-U site in the presence of both NusA and P7²⁵.”

Comment 6:

It is interesting that p7 is expected to prevent transcription initiation by cellular RNAP from all types of promoters but at the same time to act as an antiterminator, suggesting that some RNAP should still be engaged in RNA synthesis – how these two activities can be reconciled in an integrated model of transcription regulation?

The authors say in their reply that transcription of late phage genes could be performed by the phage-encoded RNAP when the host is totally shut down by P7. However, this does not explain how P7 could act as an antiterminator if the host RNAP is inhibited – it is only possible if RNAP can initiate transcription.

Reply:

Konstantin Severinov’s lab has done wonderful works to understand the regulation of Xp10 phage gene expression [Semenova *et al.*, 2005, PMID 15661002; Yuzenkova *et al.*, 2003, PMID: 12850143]. However, many details are still not fully understood. They have found that host RNAP is solely responsible for transcription of Xp10 phage early genes, but the late genes of Xp10 are transcribed by both host RNA polymerase and Xp10 RNAP [Semenova *et al.*, 2005, PMID 15661002; Yuzenkova *et al.*, 2003, PMID: 12850143]. The data suggested that P7 and Xp10 RNAP together serve as the switch to turn on expression of phage late genes, but it is unclear whether they function redundantly or coordinately.

As pointed by the review, it seems that P7 is not able to act as an antiterminator if it inhibits transcription initiation. However, P7 is still able to turn on expression of late genes under certain circumstances. For example, in the early stage of infection when P7 molecules are far less than host RNA polymerase, P7 might engage with the elongated RNAP and consequently turns on the expression of late genes.